# Engineering Af1521 improves ADP-ribose binding and identification of ADP-ribosylated proteins

Kathrin Nowak[1,2], Florian Rosenthal[1], Tobias Karlberg[3], Mareike Bütepage[4], Ann-Gerd Thorsell[3], Birgit Dreier[5], Jonas Grossmann[6,7], Jens Sobek[6], Ralph Imhof[1], Bernhard Lüscher [4], Herwig Schüler [3], Andreas Plückthun [5], Deena M. Leslie Pedrioli [1] & Michael O. Hottiger [1✉]

Protein ADP-ribosylation is a reversible post-translational modification that regulates important cellular functions. The identification of modified proteins has proven challenging and has mainly been achieved via enrichment methodologies. Random mutagenesis was used here to develop an engineered Af1521 ADP-ribose binding macro domain protein with 1000-fold increased affinity towards ADP-ribose. The crystal structure reveals that two point mutations K35E and Y145R form a salt bridge within the ADP-ribose binding domain. This forces the proximal ribose to rotate within the binding pocket and, as a consequence, improves engineered Af1521 ADPr-binding affinity. Its use in our proteomic ADP-ribosylome workflow increases the ADP-ribosylated protein identification rates and yields greater ADP-ribosylome coverage. Furthermore, generation of an engineered Af1521 Fc fusion protein confirms the improved detection of cellular ADP-ribosylation by immunoblot and immuno-fluorescence. Thus, this engineered isoform of Af1521 can also serve as a valuable tool for the analysis of cellular ADP-ribosylation under in vivo conditions.

[1] Department of Molecular Mechanisms of Disease, University of Zurich, Zurich, Switzerland. [2] Molecular Life Science PhD Program of the Life Science Zurich Graduate School, University of Zurich, Zurich, Switzerland. [3] Department of Biosciences and Nutrition, Karolinska Institute, Huddinge, Sweden. [4] Institute of Biochemistry and Molecular Biology, RWTH Aachen University, Aachen, Germany. [5] Department of Biochemistry, University of Zurich, Zurich, Switzerland. [6] Functional Genomics Center Zurich, ETH Zurich and University of Zurich, Zurich, Switzerland. [7] SIB Swiss Institute of Bioinformatics, Quartier Sorge - Batiment Amphipole, Lausanne, Switzerland. ✉email: michael.hottiger@dmmd.uzh.ch

ADP-ribosylation is a ubiquitous and reversible post-translational modification (PTM) found in viruses, bacteria, and eukaryotes[1,2]. The attachment is catalyzed by members of the ADP-ribosyltransferase (ART) family, which transfer ADP-ribose (ADPr) from nicotinamide adenine dinucleotide (NAD$^+$) onto proteins. Specifically, ADPr is transferred onto amino acids containing nucleophilic oxygen, nitrogen, or sulfur side chains, which results in N-, O-, or S-glycosidic-ribose linkages[2]. ARTs are often found as multi-domain proteins[3,4]. In addition to their catalytic domains, additional domains are suggested to regulate interactions with other macromolecules and the specificity of ADP-ribosylation activities. Most ARTs modify proteins, but some are known to ADP-ribosylate other target molecules, including tRNAs[5,6], polynucleic acids[7,8], and small-molecule antimicrobials[9,10]. ADP-ribosylation comes in two flavors: as mono-ADP-ribosylation (MARylation) and as poly-ADP-ribosylation (PARylation).

Studies have demonstrated that protein-linked ADPr modifications are capable of both hydrogen bonding and hydrophobic interactions that alter enzymatic activities, interactions with binding partners, and regulate the stability of modified proteins[11–13]. ADP-ribosylation also functions as a signaling scaffold for the recruitment of binding proteins to ADPr-modified targets[1,14]. Several different protein motifs and domains able to bind poly-ADPr (e.g., poly(ADP-ribose) (PAR)-binding motif), two consecutive ADPr units (e.g., PAR-binding zinc finger) or iso-ADPr (e.g., WWE domain) are known, and referred to as ADP-ribosylation "readers"[15]. In addition, macro domains can recognize the terminal ADPr unit and bind to monomeric ADPr (MAR) and its derivatives, including ADP-ribose-1″-phosphate, O-acyl-ADP-ribose, and the terminal ADP-ribose of PAR, as well as protein-conjugated MAR or PAR (MARylated or PARylated proteins)[15–17]. Two protein families, the ADP-ribosyl-acceptor hydrolases and the macro domain-containing enzymes, have diverse target specificity and hydrolytic activities toward proteins modified with ADPr and, as such, function as ADPr modification erasers[1,14,18,19].

To further elucidate the cellular function of ADP-ribosylation, it is of importance to identify the ADP-ribosylome—the modified proteins and their ADPr acceptor amino acids present in the sample under investigation. This is, however, challenging due to their low abundance and dynamic turnover compared to other PTMs[20,21]. Although protein ADP-ribosylation was first described in the early 1960s (ref. [22]), ADP-ribosylation was traditionally studied and identified in vitro via the incorporation of radioactive ADPr or ADPr-analogs. For a long time, only antibodies recognizing PAR were available, which restricted our ability to detect only PAR events by immunoblotting or immunofluorescence (IF). Only recently ADPr-binding domains, like the macro domain of the *Archaeoglobus fulgidus* Af1521 fused to an Fc fragment or anti-ADPr antibody that can detect mono-ADP-ribosylation, became available[23].

Originally, ADP-ribosylation was studied as a nuclear PTM written by ARTD1 (also known as PARP1) following genotoxic stress induced by hydrogen peroxide (H$_2$O$_2$)[24]. This changed with the development of protein mass spectrometry (MS)-based tools that facilitated analysis of this, and other, dynamic PTMs[25–27]. The large amount of experimental data that has been generated and the technical progress that has been made over the past decade has significantly advanced our understanding of the function of ADP-ribosylation on the molecular level[1,21,28,29].

Among the different ADPr-binding domains, the macro domain is a conserved protein fold that exists either as a single-domain protein or as part of a larger protein, which has been identified in viruses, bacteria, archaea, and eukaryotes (reviewed in refs. [30,31]). Originally identified as X-domains in viruses[32],

these conserved regions were renamed macro domains due to their similarity to the C-terminal domain of the histone H2A variant called MacroH2A[33]. Twelve macro domain-containing proteins are encoded by the human genome[2,19,30] and recent bioinformatic analyses have subclassified them into six groups[31]. This protein domain typically consists of 130–190 amino acids that adopt a distinct fold consisting of a central beta sheet surrounded by four to six helices[34]. The structurally characterized macro domain, Af1521[16] has been used to affinity-purify ADP-ribosylated proteins[35] or to enrich ADP-ribosylated peptides for subsequent MS analysis to identify ADP-ribosylated proteins and localize ADPr acceptor site[26]. Enrichment efficiencies were, however, limited by the low affinity of the macro domain toward ADP-ribosylated peptides.

A possible way to overcome this limitation would be to engineer a variant of the Af1521 macro domain with higher affinity for ADPr. Directed in vitro evolution of proteins has, indeed, become a widely applied strategy to generate protein variants with a desired property[36]. Especially for the generation of high-affinity binders, the iterative succession of randomization and selection was shown to efficiently mimic natural affinity maturation[37,38]. The success of this approach is dependent on the size and the quality of the library, and can employ methods that work entirely in vitro, like ribosome display[39] and mRNA display[40] methodologies. Ribosome display selection allows for the selection of binders with improved properties from very large libraries reaching $10^{12}$ in complexity and, because of its in vitro nature, facilitates an iterative process of randomization and selection. Randomization occurs either at a low rate by the intrinsic error rate of the polymerase used or can be enhanced using error-prone PCR amplification methods, DNA shuffling, or both; thereby generating highly diverse pools[41–44].

Here, we increase the affinity of the Af1521 macro domain for ADPr by 1000-fold ($K_D$ from ~3 μM to ~3 nM) using random mutagenesis and in vitro ribosome display selection using an ADP-ribosylated histone peptide. The crystal structure of the engineered Af1521 (eAf1521) macro domain provides an explanation for the increased affinity. Comparison of the wild-type (WT) and eAf1521 macro domains using our proteomic ADP-ribosylome approach reveals an increase in the identification rates of modified peptides present in both untreated and H$_2$O$_2$-treated HeLa cell lysates, with the eAf1521 macro domain. Finally, fusion of the eAf1521 macro domain with an Fc fragment expands the application of this tool for classic molecular biology applications. Further characterization of an Fc fragment-containing eAf1521 confirms the improved detection of ADP-ribosylation by immunoblotting and IF, thus allowing the detection of additional ADP-ribosylation events.

## Results

**In vitro selection of engineered Af1521 with increased affinity.** To decipher cellular ADP-ribosylomes and identify the corresponding ADPr acceptor sites, we recently codeveloped a MS-based approach that employs the Af1521 macro domain to enrich ADP-ribosylated peptides from different sources[45,46]. To improve the binding affinity of WT Af1521 for ADPr and enhance the detection of ADP-ribosylated peptides or proteins, affinity maturation of WT Af1521 was performed by a combination of error-prone PCR and in vitro ribosome display selection (Supplementary Fig. 1a). This was done using an H2B peptide that was synthetically ADP-ribosylated with a N-glycosidic linkage on Gln (Q) at position 2 of the peptide[47,48] (Supplementary Fig. 1b). After performing the error-prone PCR, the Af1521 mutants were in vitro transcribed, translated (such that they do not leave the ribosome[38]) and subsequently selected for binding

to the ADP-ribosylated peptide (Supplementary Fig. 1b). After every selection round, the enriched mRNA was isolated and amplified by RT-PCR. The enriched pools were subcloned into the ribosome display vector pRDV which served as template for the next round of selection[38,48,49]. Four rounds of selection using the modified peptide coupled either to streptavidin-coated plates (for rounds 1–3) or magnetic beads (for round 4) were performed with increasing stringency. This was achieved by extending the washing times in rounds 2, 3, and 4, reducing the target concentration from 200 nM ADPr H2B peptide to 100 nM in round 2, and to 20 nM in rounds 3 and 4. Off-rate selection was also implemented using unmodified H2B peptide for round 3 and auto-ADP-ribosylated ARTD10 for round 4 (refs. [50,51]).

After the fourth ribosome display selection iteration, the pools were subcloned into an *Escherichia coli* expression vector for subsequent analysis of the mutated Af1521 domains by ELISA, using the modified H2B peptides and its unmodified counterpart[52]. A total of 88 out of 92 clones (>95%) displayed specific and significant ADP-ribosylated peptide-binding properties (binding signals > 0.5 absorbance units) compared to the unmodified peptide (Supplementary Fig. 1c). Due to the large number of positive binders, we randomly chose ten clones and subcloned them as GST-fusion proteins. After expression and purification, increasingly stringent pull-down assays using unmodified and ADP-ribosylated H2B peptides were performed to further characterize the candidate binders (Fig. 1a and Supplementary Fig. 1d). While binding of WT Af1521 to the ADP-ribosylated peptide was compromised at salt concentrations of 200 mM and lost at 400 mM, we observed that only one of the tested macro domain candidates, which we termed eAf1521, was still able to bind the modified peptide very well in the presence of 400 mM NaCl. Sequencing revealed that this eAf1521 variant contains nine point mutations (Fig. 1b). Based on the available structure of WT Af1521, we noticed that one of these mutations (Y145R) occurs within the ADPr-binding region of the macro domain[45].

In addition, pull-down experiments with either GST-WT Af1521 and GST-eAf1521, and the modified H2B peptides in the presence of increasing concentrations of free ADPr as competitor confirmed the strong binding of eAf1521 compared to WT Af1521 (Supplementary Fig. 1e). Low concentrations of free ADPr completely inhibited WT Af1521 binding to the ADP-ribosylated peptide, while eAf1521 ADP-ribosylated peptide binding was still observed even in the presence of 10× more free ADPr relative to the modified peptide concentration. These findings demonstrate a strong improvement of the binding affinity of eAf1521 for the ADP-ribosylated H2B peptide compared to WT Af1521.

To quantify the binding affinities of WT Af1521 and eAf1521 to ADPr, we performed surface plasmon resonance (SPR) measurements. For these experiments, recombinant His-tagged WT Af1521 or eAf1521 were expressed and purified, and then immobilized on a (multi)NTA-modified NiHC1000M chip and their affinities for soluble ADPr analyzed (Supplementary Fig. 1f). These kinetic analyses revealed that the $K_D$ of WT Af1521 for ADPr was ~3 μM, while eAf1521 displayed a $K_D$ of ~3 nM (Table 1), and thus a ~1000-fold affinity increase.

As the Af1521 macro domain also exhibits hydrolytic activity that could compromise modified protein detection and enrichment potentials, we compared the catalytic activities of WT Af1521 and eAf1521 using an in vitro de-ADP-ribosylation assay. To this end, the catalytic domain of ARTD8 (ARTD8cat) was auto-modified using radiolabeled $^{32}$P-NAD$^+$, residual NAD$^+$ removed, and the resulting mono-ADP-ribosylated ARTD8cat used to define the hydrolytic activities of WT Af1521 or eAf1521 (Supplementary Fig. 1g). While both tested enzymes remained

inactive at 4 °C even after 2 h incubation, the condition used for ADP-ribosylome enrichment for MS analysis, both exhibited hydrolase activity at 37 °C and completely demodified ARTD8cat during the 2 h incubation period. Together, these data demonstrate that the mutations introduced into eAf1521 significantly increase ADPr-binding affinities but do not abolish macrodomain ADPr hydrolase activity.

**K35E and Y145R enhance binding affinity in eAf1521.** To gain further mechanistic insights into the higher affinity of eAf1521 for ADPr, we studied the structural changes introduced by the nine mutations. To this end, we solved the crystal structure of eAf1521 in a complex with ADPr at 1.82 Å resolution (Table 2). The overall structure of the eAf1521 macro domain was virtually identical to WT Af1521 (Fig. 1c). The two models align with a root mean square difference of 0.3 Å (calculated over all 192 shared Cα positions)[53]. Nevertheless, two notable features could explain the improved affinity of eAf1521 for ADP-ribosylated targets. First, the orientation of the proximal ribose moiety of eAf1521-bound ADPr was rotated such that the oxygen at the C-1″, which serves as the anchor point for the amino acid acceptor side chain, resided closer to the macro domain surface (Fig. 1d). This change was facilitated by the K35E and Y145R amino acid substitutions. The R145 side chain forms a salt bridge with the E35 carboxylate, which provides a rigid boundary for proximal ribose binding, while allowing electrostatic interaction between the Arg Nε atom and the C-4′ (ring-forming) oxygen. In the WT Af1521 structure, the proximal ribose site is delineated by the Y145 side chain, which is situated between the ribose carbon ring on one face and the I102 side chain on the other. Indeed, this eAf1521 amino acid arrangement appears to contribute directly to strengthening the interaction with the proximal ribose. The electrostatic interaction between R145 of eAf1521 and the bound proximal ribose also changes the rotamer of the I144 side chain. In the eAf1521 structure, the I144 side chain adopts a rotamer that bridges the central phosphates of ADPr and appears to trap the ADPr (Fig. 1d and Supplementary Fig. 1h), but may not energetically contribute to the binding.

To confirm the importance of the generated salt bridge between R145 and E35, we replaced the R145 of eAf1521 with a shorter Lys residue (R145K), which lacks the Nε atom H-bonding to the C-4′ (ring-forming) oxygen. Binding experiments using our ADP-ribosylated H2B peptide revealed that the GST-eAf1521-R145K mutant was no longer able to bind the ADP-ribosylated peptide, comparable to a known nonbinding Af1521 macro domain mutant (Af1521-G42E[35]; Supplementary Fig. 1i). These data confirmed the importance of the R145 residue and demonstrated that the positively charged Lys by itself was not sufficient to maintain the high-affinity binding of ADPr peptides and revealed that the guanidino group is essential for the contact to both E35 and the ring-forming C-4′ oxygen.

To further test whether E35 and R145, and not the other seven mutations, are the main contributors to the observed increase in affinity of eAf1521 for ADPr, we introduced K35E and Y145R into WT Af1521 and vice versa (Fig. 1e). Binding assays confirmed that the GST-eAf1521-E35K-R145Y mutant lost its increased affinity and now bound ADP-ribosylated peptides comparable to WT Af1521. In support, GST-WT Af1521-K35E-Y145R displayed significantly augmented affinity for the ADP-ribosylated H2B peptide (Fig. 1e), which became comparable to eAf1521 (Fig. 1a).

Finally, to define the influence that I144 tunnel formation covering the ADPr pyrophosphate had on eAf1521 ADPr binding, we generated I144G eAf1521 mutants and performed additional binding assays, as described above (Fig. 1e). The

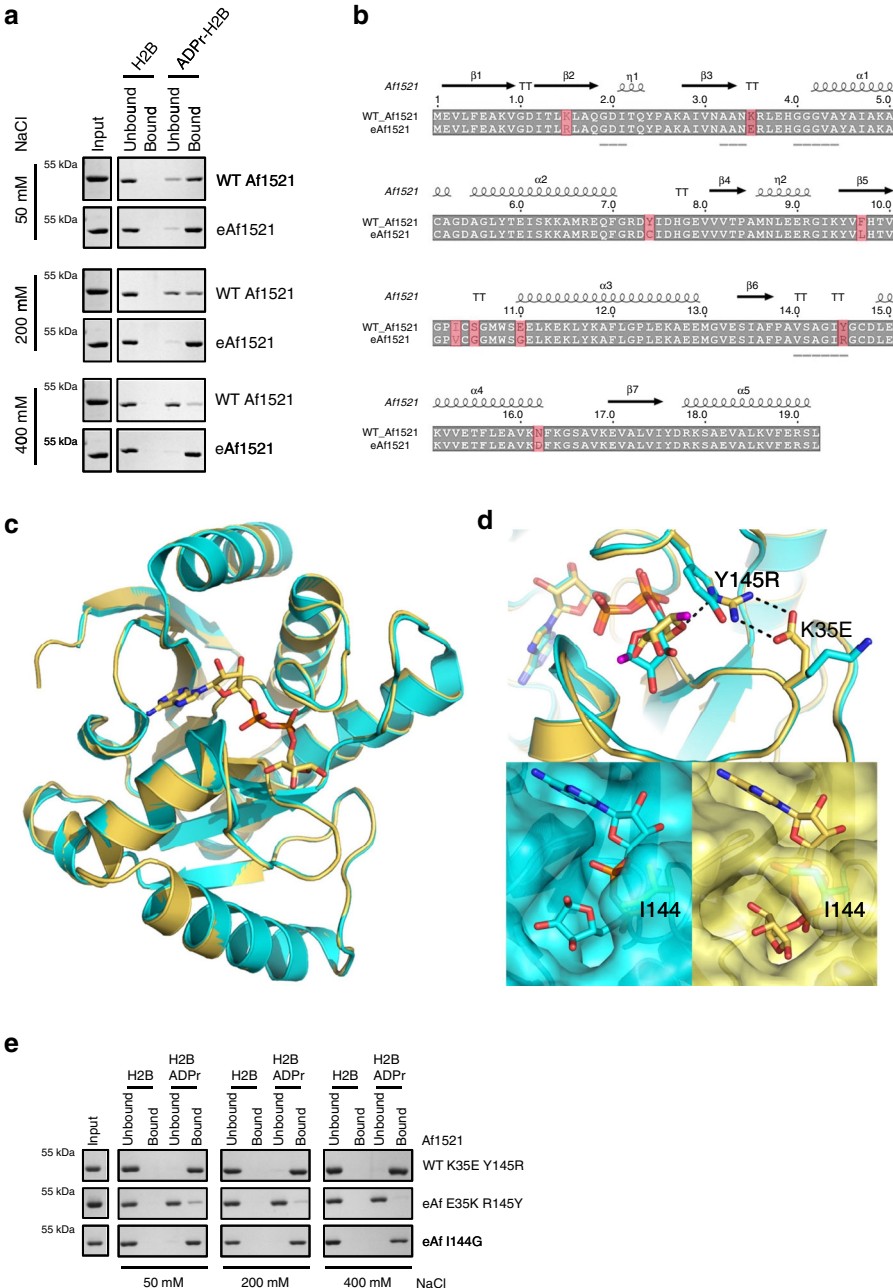

**Fig. 1 Characterization of engineered Af1521 macro domain. a** Pull-down experiment using unmodified and ADP-ribosylated H2B peptides (biotinylated and bound to streptavidin Sepharose beads) of WT Af1521 and eAf1521 under conditions with increasing salt concentrations. Input lanes are the same for all measured conditions. The proteins were detected with SDS–PAGE followed by Coomassie staining. The experiment was repeated independently ($n = 3$) with similar results. **b** Sequence alignment of WT Af1521 and eAf1521. Point mutations in eAf1521 are highlighted. The binding region is underlined. **c** Structural comparison of the eAf1521 (mustard) and WT Af1521 (turquoise; PDB ID: 2BFQ). **d** Upper panel: detailed representation of the proximal ribose of the ADPr with the point mutations Y145R and K35E in its vicinity. The C-1′-linked oxygen is shown in purple. Lower panel left: surface rendering of the WT Af1521 ADPr-binding site. The I144 side chain is highlighted. Lower panel right: corresponding view for the eAf1521 macro domain, with the I144 side chain rotated to entrap the diphosphate moiety. **e** Pull-down experiment using unmodified and ADP-ribosylated H2B peptides (biotinylated and bound to streptavidin Sepharose beads) of WT Af1521-K35E-Y145R, eAf1521-E35K-R145Y, and eAf1521-I144G under increasing salt concentration. Input lanes are the same for all measured conditions. The proteins were analyzed by SDS–PAGE followed by Coomassie staining. The experiment was repeated independently ($n = 2$) with similar results. Source data are provided as a Source data file.

binding of eAf1521-I144G remained similar compared to eAf1521 indicating that I144 and the formation of the tunnel did not contribute to the increased binding we observed with eAf1521. Together, these results provide strong evidence that the two mutations K35E and Y145R and, as a consequence, the generated salt bridge solely contribute to the increased binding affinity of eAf1521.

**Enhanced enrichment of ADP-ribosylation using eAf1521.** The experiments described above were all carried out on free ADPr or a synthetically ADP-ribosylated H2B peptide. To investigate whether the in vitro selection procedure could improve our in vivo ADP-ribosylome MS analysis pipeline, we compared the HeLa cell $H_2O_2$-induced ADP-ribosylated peptide enrichment capacities of WT Af1521 and eAf1521, using our established MS

**Table 1 Kinetic data for binding of ADPr to eAf1521 and WT Af1521, respectively.**

| Ligand | Ligand density | Model | $k_{on}/10^5\,M^{-1}s^{-1}$ | $k_{off}/s^{-1}$ | $K_D/nM$ | $\chi^2$ | RU (max) |
|---|---|---|---|---|---|---|---|
| WT Af1521 | 4650 | 1:1 | 2.36 ± 0.02 | 0.678 ± 0.003 | 2879 ± 9 | 0.028 | 33.3 |
| | | ss | | | 9710 ± 1950 | 0.068 | 93.2 |
| eAf1521 | 520 | 1:1 | 5.94 ± 0.02 | 0.001115 ± 0.000002 | 1.88 ± 0.01 | 0.042 | 9.6 |
| eAf1521 | 500 | 1:1 | 5.63 ± 0.06 | 0.001203 ± 0.000007 | 2.14 ± 0.01 | 0.041 | 10.2 |
| eAf1521 | 390 | 1:1 | 5.15 ± 0.04 | 0.001000 ± 0.000006 | 1.94 ± 0.01 | 0.032 | 9.5 |

A dissociation rate constant, $k_{off}$, was calculated by using $K_D$ from steady-state measurement and the association rate constant, $k_{on}$. Statistical error from fitting.
ss steady state.

**Table 2 Data collection and refinement statistics.**

| | eAf1521:ADP-ribose |
|---|---|
| Data collection | |
| Space group | C2 |
| Cell dimensions | |
| $a, b, c$ (Å) | 141.31, 39.54, 35.96 |
| $\alpha, \beta, \gamma$ (°) | 90, 97.704, 90 |
| Resolution (Å) | 70.02–1.82 (1.85–1.82)* |
| $R_{merge}$ | 0.185 (2.528) |
| $I/\sigma(I)$ | 8.6 (1.4) |
| Completeness (%) | 99.7 (96.0) |
| Redundancy | 6.4 (5.4) |
| Refinement | |
| Resolution (Å) | 1.82 |
| No. reflections | 17778 |
| $R_{work}/R_{free}$ | 0.187/0.212 |
| No. atoms | |
| Protein | 1476 |
| Ligand/ion | 36 |
| Water | 100 |
| B-factors | |
| Protein | 27.0 |
| Ligand/ion | 25.0 |
| Water | 30.5 |
| R.m.s. deviations | |
| Bond lengths (Å) | 0.01 |
| Bond angles (°) | 1.02 |

*All values based on one crystal. Values in parentheses are for highest-resolution shell.

work flow[26,54] (Supplementary Fig. 2a). This well-established workflow was applied to both untreated and $H_2O_2$-treated HeLa cell lysates with either WT Af1521 (Supplementary Fig. 2b, c) or eAf1521 (Supplementary Fig. 2d, e). Comparison of the untreated and $H_2O_2$-treated HeLa cell lysates confirmed a strong increase of ADP-ribosylation on both the peptide (Supplementary Fig. 2b, d) and protein (Supplementary Fig. 2c, e) levels, presumably via $H_2O_2$-induced activation of nuclear ARTD1 (Supplementary Data 1 and 2). Interestingly, eAf1521 enriched significantly more ADP-ribosylated peptides from both untreated and $H_2O_2$-treated HeLa cell lysates (Fig. 2a, b). In the untreated HeLa samples, ADP-ribosylation levels are typically not detectable by conventional anti-PAR antibody IF[54]. Indeed, >6 times more ADP-ribosylated proteins were detected in untreated HeLa lysates with eAf1521 compared to WT Af1521 (56 compared to 9 ADP-ribosylated proteins; Fig. 2c, left panel). In agreement with this, the number of distinct ADP-ribosylated proteins identified in $H_2O_2$-treated HeLa cell lysates was twofold greater, when the enrichments were performed with eAf1521 rather than WT Af1521 (419 vs. 185 ADP-ribosylated proteins; Fig. 2c, right panel). Together, these data indicate that the higher enrichment efficiency of eAf1521 resulted in an increase in the number of modified proteins identified within a complex biological sample. Comparing the overlap between the ADP-ribosylomes identified using WT Af1521 or eAf1521, we identified >90% of all of the untreated and $H_2O_2$-treatment-induced ADP-ribosylated proteins with WT Af1521 using eAf1521 for the enrichment (Fig. 2c, right panel). The remaining ADP-ribosylated proteins identified only with WT Af1521 lie within the run-to-run variations that are typically observed during MS analyses.

To further exclude that the selection strategy of eAf1521 resulted in any bias in the enrichment of ADPr-modified proteins, comparison of our eAf1521 ADP-ribosylome with another ADPr proteomic study using WT Af1521 for enrichment of $H_2O_2$-treated HeLa lysate was performed. This comparison revealed a high-degree overlap of ADP-ribosylated proteins, strongly implying that the selection strategy did not result in any apparent bias (Supplementary Fig. 2f)[27,55]. Together, these results suggest that the observed eAf1521 affinity enhancements for ADPr are not specific to the modified H2B peptide used during selection and, instead, demonstrate that eAf1521 ADP-ribosylated peptide enrichment capacities remain ubiquitous for this PTM. To investigate the ADPr acceptor sites, we restricted our searches of EThcD spectra to the potential acceptor sites S, R, K, D, E, and Y, as the evidence provided by Hendriks et al.[27] indicated that other ADPr acceptor amino acids (e.g., C, H, and T) are not very abundant in either untreated and $H_2O_2$-treated Hela cells[27]. In agreement with these findings, ADPr acceptor site analysis revealed that Ser (S) was the major modified amino acid identified in the HeLa $H_2O_2$-treated samples[54,56] (Fig. 2d and Supplementary Data 3).

The current MS-based workflows employed by the community for the identification of cellular ADP-ribosylome relies on large amounts of input material[26,54]. This has remained a major limitation with respect to the type of samples that are feasible for ADP-riboylome analyses and, thus, has limited our potential ability to analyze clinical samples (e.g., patient biopsies). To investigate whether eAf1521-based MS analysis could reduce starting material requirements, we used the same workflow with 5, 10, or 20 mg of $H_2O_2$-treated HeLa lysate as input material and carried out label-free quantification (LFQ) analyses. Three independent MS sample measurements identified similar amounts of ADP-ribosylated proteins (Supplementary Fig. 2i and Supplementary Data 4). Comparison of the identified ADP-ribosylated proteins enriched by eAf1521 with the corresponding lysate amounts revealed that the procedure was reproducible, and with all three tested concentrations the number of identified ADP-ribosylated proteins increased, when enrichments were carried out with eAf1521 (Fig. 2e). A total of 20 or 10 mg identified comparable numbers of modified proteins and, importantly, reducing the amount of input material to 5 mg still facilitated the identification of 80% of the ADP-ribosylated proteins identified with the maximal input material, implying that the highest abundant ADP-ribosylated proteins are still detectable (Fig. 2f). Together, these data suggest that valuable

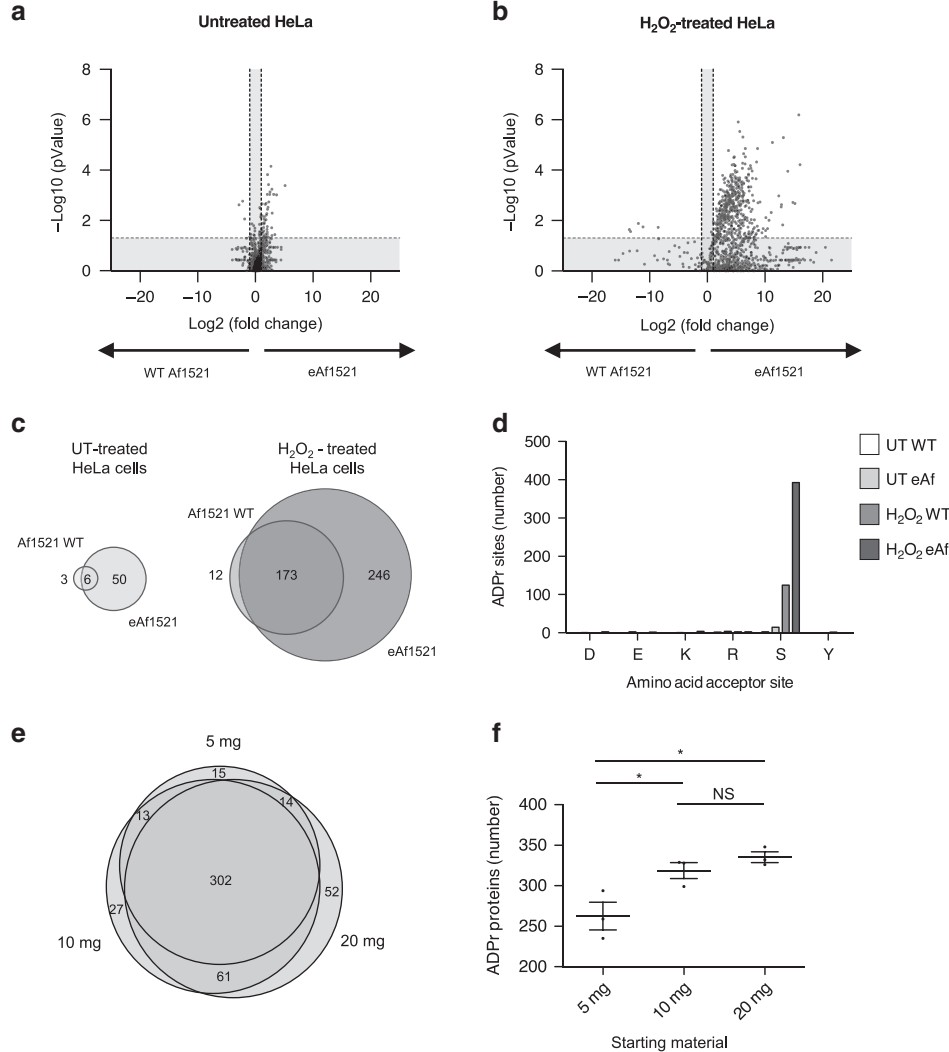

**Fig. 2 Stronger enrichment of ADP-ribosylome by eAf1521 compared to WT Af1521. a, b** Volcano plots comparing ADP-ribosylated peptides enriched from untreated and $H_2O_2$-treated HeLa cells, with either WT Af1521 or eAf1521. The statistical analysis and $p$ value calculation was performed with two-tailed Student's $t$ test, the black lines represent foldchange > 2 and $p$ value < 0.05. **c** Venn diagrams displaying the overlap of ADP-ribosylated protein groups enriched from untreated (left panel) or $H_2O_2$-treated HeLa cells (right panel), with either WT Af1521 or eAf1521. **d** Distribution of ADPr on different amino acid acceptor sites (D, E, K, R, S, and Y). Only EThcD data with a site localization score of 95% were considered. **e** Venn diagram representing the overlap of ADP-ribosylated protein groups enriched with eAf1521 using different amounts of starting material (5, 10, and 20 mg). **f** Scatter plot comparing the number of ADP-ribosylated protein groups identified via eAf1521 enrichment using different amounts of starting material (5, 10, and 20 mg). Data are presented as mean ± SEM ($n = 3$ biochemical independent samples). The statistical analysis and $p$ value calculation was performed with two-tailed Student's $t$ test (*$p < 0.05$, ns not significant). Source data are provided as a Source data file.

ADP-ribosylome data can be acquired from smaller amounts of input material using eAf1521-based enrichments, which represents an important advancement and may prove valuable for the analysis of disease and/or clinical samples.

**Improved ADPr peptide and ADPr acceptor site identification.** When analyzing the MS signal intensities of all ADP-ribosylated peptides, we observed strong increases in the intensities in $H_2O_2$-treated HeLa cell lysates compared to the untreated condition (Fig. 3a). Furthermore, global ADP-ribosylated peptide intensities were enhanced independent of the condition (untreated vs. $H_2O_2$-treated) when the enrichment was carried out with eAf1521. Ultimately, this led to significant increases in the number and purity of ADP-ribosylated peptide spectra matches (PSMs) identified with eAf1521compared to WT Af1521 (Supplementary Fig. 3a, b).

To further investigate peptides with different ADPr acceptor sites and define the potential contribution of either WT Af1521 or eAf1521, we selected ADP-ribosylated peptides that were modified on either Ser (S) or Arg (R) residues with a site localization probability >95% (Fig. 3b, c). We observed that MS signal intensities generally increased for S-ADPr-modified peptides after treatment with $H_2O_2$. These S-ADPr-modified peptides belong to proteins that localize to the nucleus and are most likely targets of $H_2O_2$-activated ARTD1. Interestingly, some S-ADPr-modified peptides shown here (ARTD1/PARP1, SKGQVKEEGINKSEK, S507; HNPNPA1, SSGPYGGG GQYFAPR, S337) were also identified in the untreated HeLa cell lysates, suggesting that ARTD1 regulates cellular processes via ADP-ribosylation also under basal conditions, using the same modification sites. In contrast, the intensities of the R-ADPr-modified peptides detected here did not

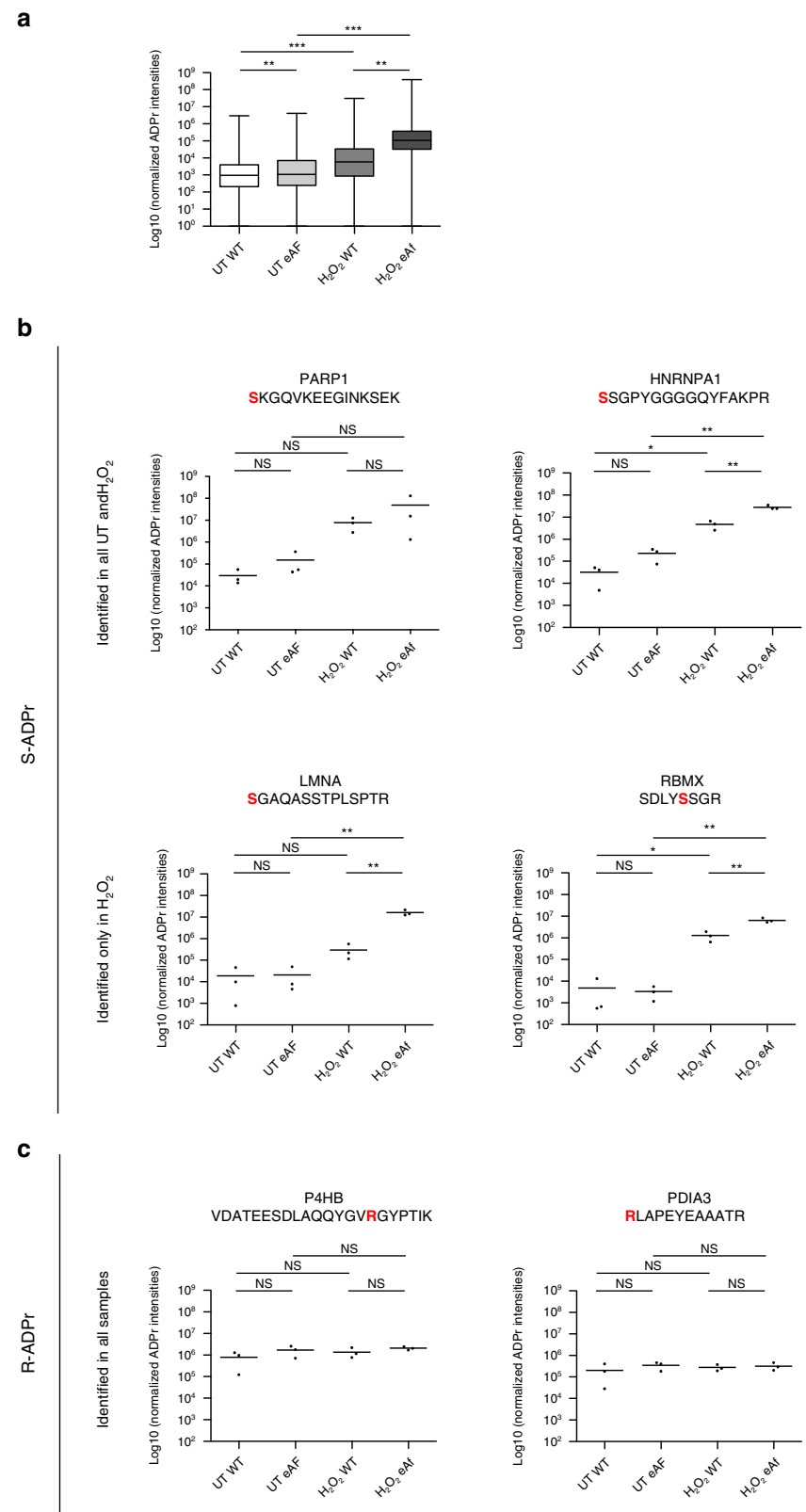

change after $H_2O_2$ treatment (Fig. 3c). Moreover, the R-ADPr-modified proteins localize to the cytoplasm, suggesting that these targets are not modified by ARTD1 and are not affected by $H_2O_2$ treatment.

In addition, we compared the ability of WT Af1521 and eAf1521 to enrich S-ADPr- vs. R-ADPr-modified peptides. Most of the S-ADPr-modified peptide intensities were significantly enhanced using eAf1521 compared to WT Af1521, which was not the case for the R-ADPr-modified peptides (Fig. 3b, c and Supplementary Fig. 3c). Nevertheless, eAf1521 was still capable of enriching R-ADPr-modified peptides to the same relative extent as WT Af1521.

**Fig. 3 eAf1521 strongly enriches for Ser-ADP-ribosylation. a** Box plot representing MS1 intensities of all ADPr-modified peptides of untreated and $H_2O_2$-treated HeLa cell lysate enriched with either WT Af1521 or eAf2151. Box extends from the 25th to 75th percentiles, the band indicates the median and whiskers extend to min and max ($n = 3$ biochemical independent samples). The statistical analysis and $p$ value calculation was performed with two-tailed Student's $t$ test (**$p < 0.01$, **$p < 0.001$). **b** Scatter plots depicting MS1 ADPr-modified peptide intensities of selected S-ADPr-modified peptides. The modification confidence of the amino acid acceptor sites are >95%. Data are presented as mean ($n = 3$ biochemical independent samples). The statistical analysis and $p$ value calculation was performed with two-tailed Student's $t$ test (*$p < 0.05$, **$p < 0.01$, ns not significant). **c** Scatter plots depicting MS1 ADPr-modified peptide intensities of selected R-ADPr-modified peptides identified in all untreated and $H_2O_2$-treated HeLa lysates. The modification confidence of the amino acid acceptor sites are >95%. Data are presented as mean ($n = 3$ biochemical independent samples). The statistical analysis and $p$ value calculation was performed with two-tailed Student's $t$ test (ns not significant). Source data are provided as a Source data file.

**eAf1521 improves detection of MAR and PAR via immuno-blotting**. The increased affinity of the eAf1521 for ADPr and the subsequent increase in the identification of ADP-ribosylated proteins, we observed under all conditions in our proteomic studies prompted us to test whether eAf1521 could be used to detect cellular ADP-ribosylation by immunoblotting. To this end, WT Af1521 and eAf1521 were fused to an Fc domain of IgG2a to generate a bivalent fusion protein (Supplementary Fig. 4a). The constructs were expressed in HEK 293-T cells and purified from the supernatants using a Ni-NTA bead-based purification, exploiting the C-terminal His tag behind an HA tag on the Fc domain (Supplementary Fig. 4b).

The fusion constructs were then tested for immunoblot efficacy using three different extracts: (i) a HeLa cell lysate that was treated with $H_2O_2$ alone or together with the broad PARP inhibitor PJ34, (ii) in vitro auto-modified poly-ADP-ribosylated ARTD1 that was subsequently left untreated or treated with the enzyme poly(ADP-ribose)glycohydrolase (PARG) to reduce PARylation to mono-ADP-ribosylated ARTD1, or (iii) in vitro auto-modified mono-ADP-ribosylated ARTD8cat (Fig. 4a). In all tested conditions, eAf1521 recognized the modified proteins to a stronger extent, irrespective of whether the proteins were mono- or poly-ADP-ribosylated. Treatment of samples with PJ34 reduced the ADP-ribosylation signal after $H_2O_2$ treatment, indicating that the signals detected were indeed protein ADP-ribosylation. The same experiment was performed with different concentrations of the purified Af1521 Fc constructs (Supplementary Fig. 4c). The detected differences between WT Af1521 or eAf1521 could be confirmed at all concentrations; at the highest concentration used (500 ng/ml), eAf1521 signals were stronger than WT. Moreover, at lower concentrations, WT Af1521 signals were not detectable (125 or 31.25 ng/ml), whereas with 31.25 ng/ml eAf1521 both poly- or mono-ADP-ribosylated proteins were still detected. These findings demonstrate that eAf1521 also displays a ~16-fold increase in immunoblot detection sensitivity when compared to WT Af1521. We also tested the specificity of Fc-WT Af1521 and Fc-eAf1521 constructs by performing competition experiments with free ADPr against in vitro auto-modified poly-ADP-ribosylated and mono-ADP-ribosylated ARTD1. Co-incubation of our Fc-proteins with free ADPr reduced the signal on immunoblots, confirming the specificity toward ADP-ribosylated proteins (Supplementary Fig. 4d).

To specifically test oligo-ADP-ribosylation detection sensitivity, we auto-modified ARTD1 in vitro using different $NAD^+$ concentrations to generate either oligo- or poly-ADP-ribosylated ARTD1. Subsequent immunoblot analysis confirmed that eAf1521 detects both oligo- and poly-ADP-ribosylated ARTD1 (Supplementary Fig. 4e). Next, we aimed to determine the enrichment capability of eAf1521 toward poly-ADP-ribosylated proteins. Therefore, we performed poly-ADP-ribosylated ARTD1 pull-down assays using our GST-fusion constructs (Supplementary Fig. 4f). Poly-ADP-ribosylated ARTD1 was enriched using both WT Af1521 and eAf1521 GST-fusion proteins compared to the control of GST alone. Finally, we tested whether eAf1521 also

detects free PAR chains that were isolated from poly-ADP-ribosylated ARTD1 digested with proteinase K. Dot blot analysis revealed that Fc-eAf1521 indeed also recognized isolated PAR chains more efficiently than Fc-WT Af1521 (Supplementary Fig. 4g). Together, these data indicate that the mutations introduced into eAf1521 increased its affinity toward ADPr, without altering the preference for mono-, oligo-, or poly-ADP-ribosylated targets.

**eAf1521 reveals extranuclear basal ADP-ribosylation**. The increased sensitivity observed in immunoblotting led us to investigate whether the same constructs would also show greater sensitivity when used for IF detection of cellular ADP-ribosylation in cells. To this end, HeLa cells were left untreated or treated with $H_2O_2$ for 10 min, fixed with 4% PFA, and, subsequently, incubated with Fc-WT Af1521 or Fc-eAf1521 (Fig. 4b). Treatment with $H_2O_2$ led to stronger signals of nuclear ADP-ribosylation when detected with eAf1521 compared to WT Af1521 (Fig. 4b). When subtracting the ADPr WT Af1521 signal from the same condition detected by eAf1521, the overall enhancement was still evident (Supplementary Fig. 5a), confirming the enhanced ability of eAf1521 to detect cellular ADP-ribosylation.

To further investigate the specificity of our generated tool toward ADP-ribosylation, we pretreated HeLa cells with the ADP-ribosylation inhibitor olaparib[57] to inhibit ARTD1. Inhibition of ARTD1 eliminated the nuclear signal detected in $H_2O_2$-treated HeLa cells with both WT Af1521 and eAf1521, strongly implying that WT Af1521 and eAf1521 were specifically recognizing ADP-ribosylation (Fig. 4b). Furthermore, we performed a co-incubation experiment of WT Af1521 and eAf1521 in the presence of either 1 μM free ADPr or another nucleotide (e.g., GTP) on $H_2O_2$-treated HeLa cells (Supplementary Fig. 5b). The detected nuclear signal was reduced by co-incubation of both our Fc fusion proteins (i.e., Fc-WT Af1521 and Fc-eAf1521) with ADPr but not with GTP, further confirming ADP-ribosylation detection specificity. Interestingly, we also observed weak nuclear and extranuclear signals with eAf1521 in untreated HeLa cells that were barely detectable with WT Af1521 (Fig. 4b and Supplementary Fig. 5c). In addition, treatment with olaparib strongly reduced the nuclear signal, while the extranuclear ADP-ribosylation signal was not affected, suggesting that the observed ADP-ribosylation is likely catalyzed by different ARTs.

**Discussion**

Investigating protein ADP-ribosylation is challenging due to the low abundance and dynamic nature of this PTM. Thus, the enrichment of ADP-ribosylated peptides prior to MS analysis is a prerequisite for global identification of ADP-ribosylated proteins. Our MS-based workflow successfully uses the macro domain protein Af1521 for the enrichment of ADP-ribosylated peptides. To expand ADP-ribosylome coverage, we revised the existing method by increasing the affinity of Af1521 for ADPr to improve

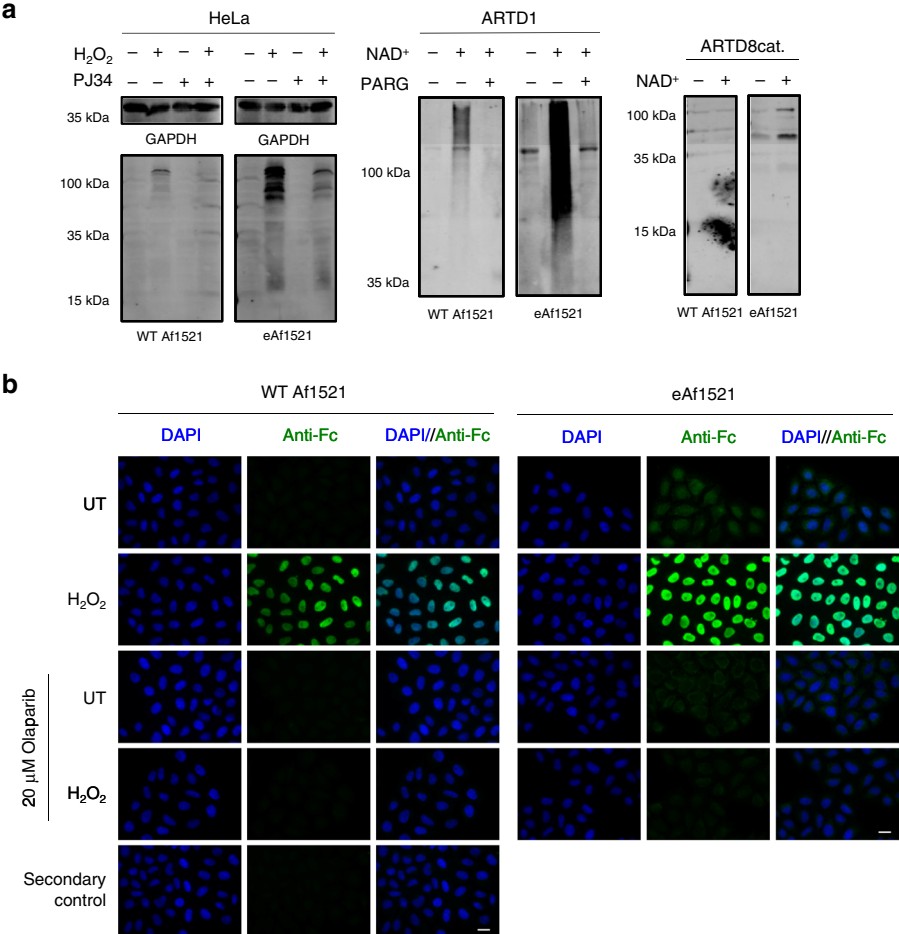

**Fig. 4 Fc-eAf1521 is more sensitive in detecting ADP-ribosylation than Fc-WT Af1521. a** Immunoblot analyses of different substrates with either Fc-eAf1521 or Fc-WT Af1521 (untreated and $H_2O_2$-treated HeLa cells, including PJ34 treatment as control for ADP-ribosylation inhibition; unmodified, poly-ADP-ribosylated and mono-ADP-ribosylated ARTD1, and unmodified and mono-ADP-ribosylated ARTD8cat). Detection was performed with IRDye 680 goat anti-mouse antibody. Immunoblot analyses to be compared were performed at the same time and with the same exposure. The experiment was repeated independently ($n = 2$) with similar results. **b** Immunofluorescence analysis of HeLa cells co-treated with $H_2O_2$ and 20 µM olaparib stained with either Fc-WT Af1521 or Fc-eAf1521, followed by Alexa 488-labeled goat anti-mouse antibody. The experiment was repeated independently ($n = 2$) with similar results. Scale bar, 20 µm. Source data are provided as a Source data file.

sample enrichment prior to MS analyses and, thus, increasing MS identification of ADP-ribosylated proteins. In addition, we also developed this engineered variant of Af1521 into a molecular tool that could be implemented with conventional molecular biology methodologies to study cellular ADP-ribosylation. By performing protein engineering and in vitro selection, we evolved a mutagenized Af1521 (eAf1521) macro domain that displays superior binding affinity for ADPr. This unique property led to the successful utilization of eAf1521 in different applications, such as peptide enrichment, immunoblotting, and IF with amplified strength.

A comparison of the crystal structures of WT Af1521 and eAf1521 revealed key conformational changes within the target-binding domain of eAf1521. Indeed, the Y145R and K35E substitutions in eAf1521 were found to be most critical. The R145 side chain forms a salt bridge with the E35 carboxylate, which constricts the motion of ADPr and forces the proximal ribose to rotate in a conformation that forms an electrostatic interaction between the Arg and the ring-forming oxygen and, in addition, brings the C-1′ oxygen closer to the domain surface, which improves access of ADP-ribosylated targets. The ribose rotation brings the C-2″ oxygen within hydrogen bonding distance from the side chain of N34 at the surface of

the domain. It is possible that a particular target protein that binds in solution might have access to additional conformations, but importantly, our studies indicate that the altered ribose conformation does not compromise target selectivity. While, the rotation of the proximal ribose induces the I144 side chain to adopt a different rotamer, our analysis shows that mutating this residue to Gly makes no energetic contribution to binding. By introducing K35E and Y145R mutation in WT Af1521, we could achieve the stronger binding toward ADPr-modified peptides as was observed with eAf1521, highlighting the contribution of these two mutations to the improved affinity. In addition, our experiments provide further evidence that the Y145 residue does not participate in catalysis even though it contributes to target binding, as the mutant Y145R still shows hydrolysis activity.

The SPR measurements revealed that the affinity of eAf1521 for ADPr was ≥1000 times higher ($K_D = $~3 nM) than that recorded for WT Af1521 ($K_D = $~3 µM). The affinity of WT Af1521 measured here was lower than that previously reported for free ADPr[16], which is likely due to the different techniques and different buffers used to define these affinities. We determined the affinity of surface-bound WT Af1521 toward ADPr using SPR, while the previous $K_D$ was measured in solution using

isothermal titration calorimetry and differences between surface and in-solution technologies have been noted before[58].

To provide further insights into the identification of ADP-ribosylated proteins and their biological functions, MS-based analysis is a powerful tool to identify PTMs[26,27,54]. The comparison of the ADP-ribosylomes enriched with either WT Af1521 or eAf1521 demonstrated that the point mutations introduced in eAf1521 improved the experimental outcomes when applied to our MS approach. Indeed, comparing the number of modified proteins enriched by either WT Af1521 or eAf1521 revealed a more than twofold increase in the number ADP-ribosylated proteins identified in $H_2O_2$-induced HeLa cells, and a sixfold increase in the number of modified proteins identified in untreated HeLa cells. The higher coverage of the ADP-ribosylome facilitated the identification of proteins that had not previously been detected using our method and, thus, will provide a deeper understanding of the cellular function of ADP-ribosylation. Comparison of different amounts of input material revealed that the amount of starting material for MS-based experiments can be reduced quite substantially. Indeed, ADP-ribosylome analysis with eAf1521 will allow us to analyze smaller tissue samples and reduce experimental costs.

Although an H2B peptide that was synthetically ADP-ribosylated with a N-glycosidic linkage on Gln at position 2 of the peptide[47] was used for the in vitro selection, ADP-ribosylated H2B peptides were not preferentially enriched with eAf1521. Almost all identified ADP-ribosylated peptides displayed higher signal intensities when eAf1521 was used for enrichment, suggesting that the eAf1521 macro domain acquired an unbiased increased affinity toward the ADPr residue.

Recently published ADP-ribosylome analyses of $H_2O_2$-treated HeLa cells highlighted that S-ADPr is the most abundant modification induced, following $H_2O_2$ activation of ARTD1[27]. We also found that S-ADPr modifications were most abundant in HeLa $H_2O_2$ samples. While, all other ADPr acceptor amino acid modified peptides were also confidently identified in the HeLa lysates analyzed, technical limitations have prohibited us from ruling out potential ADPr acceptor amino acid biases. Thus, to ensure full ADP-ribosylome enrichment/coverage, we would recommend performing the enrichment of ADP-ribosylated peptides with both WT Af1521 and eAf1521 in parallel or in tandem.

The generation of our Fc-eAf1521 fusion protein allowed us to apply the eAf1521 macro domain for use in other applications like immunoblotting and IF. Comparison of WT Af1521 and eAf1521 revealed that the evolved macro domain also has a higher binding affinity for both MARylated and PARylated proteins under denatured and fixed conditions. An earlier described construct based on WT Af1521 did not recognize ADP-ribosylation by IF in untreated HeLa cells[23], while we observed a very low ADP-ribosylation signal using our Fc-WT Af1521. This could be influenced by the different fixation method and blocking solution (methanol/acetone vs. PFA, 5% nonfat milk in PBS-T vs. 10% goat serum in PBS). Interestingly, when using eAf1521 for IF, we observed both nuclear and extranuclear ADP-ribosylation in untreated HeLa cells, suggesting that ADP-ribosylation is detectable with this tool and takes place under these conditions. This conclusion is further strengthened by the identification of ADP-ribosylated proteins in untreated HeLa cells that belong to both nuclear and extranuclear compartments. Interestingly, treatment with olaparib only eliminated the nuclear signal, while the extranuclear signal was not affected, suggesting that under these conditions ARTD1 is likely constitutively activated and only responsible for the nuclear ADP-ribosylation. Since the extranuclear signal is not detected using a conventional anti-PAR antibody[24], the nature of the extranuclear signal is

rather mono-/oligo-ADP-ribosylation. The exact localization of this ADP-ribosylation signal, as well as its potential function and importance is currently under investigations.

## Methods

**Cell culture.** HeLa cells (Kyoto, ATCC, CCL-2) and HEK 293-T (ATCC, CRL-3216) cells were cultured in Dulbecco's modified Eagle's medium (DMEM), supplemented with 10% fetal calf serum (FCS), and 1% penicillin/streptavidin at 37 °C with 5% $CO_2$. To induce ADP-ribosylation, HeLa cells were either untreated or treated with 1 mM $H_2O_2$ in PBS containing 1 mM $MgCl_2$ for 10 min. Pretreatment with ADP-ribosylation inhibitors (e.g., olaparib) was performed for 30 min before treatment with $H_2O_2$ and continued during $H_2O_2$ treatment at the indicated concentrations: PJ34 (10 μM) and olaparib (20 μM).

**Ribosome selection and screening of eAf1521 macro domain.** For the selection of Af1521 mutants with improved affinity, the *Af1521* cDNA was cloned into the pRDV plasmid backbone using the restriction enzymes BamHI and EcoRI. In order to introduce random mutations error-prone PCR was applied using the pRDV-specific primers T7B and tolAk, using either 0, 3, 6, and 10 μM of the nucleotide analogs dPTP and 8-oxo-dGTP each and Platinum Taq Polymerase (Invitrogen), as previously described[48]. For the PCR using the outer primers the following conditions applied: initial denaturation 3 min at 95 °C, for amplification 40 cycles with 30 s at 95 °C, 30 s at 50 °C, and 1 min at 72 °C followed by a final extension at 72 °C for 5 min. The four PCR reactions resulting in different mutational loads were pooled and used as template for the in vitro transcription reaction, using T7 RNA polymerase (Fermentas) and a home-made transcription buffer exactly as described[48]. The resulting RNA was purified using Illustra Microspin G-50 columns (GE). Ten μg of purified RNA were used for the in vitro translation reaction in a volume of 110 μl containing home-made S30 extract and premixZ/methionine. In vitro translation was performed at 37 °C for 15 min before the reaction was stopped as described[48]. From the stopped in vitro translation reaction four times 100 μl were used for the first selection round, only two times 100 μl for successive rounds which were pooled after the elution step. To remove unspecific binders a pre-selection step was included in rounds 2 and 3, but omitted from the initial selection round using a BSA-blocked streptavidin-coated (Immunopure, Pierce) well of a 96-well Maxisorp plate (Nunc) for 30 min at 4 °C. The reaction was then directly transferred to a fresh well additionally containing the biotinylated, streptavidin-immobilized ADP-ribosylated H2B peptide. The binding reaction was performed at 4 °C for 1 h. In the initial round, 200 nM of peptide was used, while in round 2 the target concentration was reduced to 100 nM, and in rounds 3 and 4 to 20 nM, respectively. The selection wells were washed six times (two brief washes followed by four washing steps with a 2 min incubation in round 1) with ice-cold 300 μl WTB buffer (50 mM Tris-acetate, 150 mM NaCl, 50 mM magnesium acetate, 0.05% Tween-20, pH 7.5). For all other rounds, the washing was prolonged to two fast washes and four washing steps of 10 min incubation each. Elution was performed by addition of twice 100 μl EB buffer (50 mM Tris-acetate, 150 mM NaCl, and 25 mM EDTA) followed by RNA purification and DNAseI treatment using the HighPure RNA isolation kit (Roche), as described previously[48]. Reverse transcription was performed essentially as described[48] with the exception that the scaffold-specific primer Af1521_pRDV_fwd_2 5′-GACAAAGGATCCATG-GAACGGCGTAC-3′ was used. PCR was performed using the scaffold-specific primers Af1521_pRDV_fwd_2 5′-GACAAAGGATCCATGGAACGGCGTAC-3′ and Af1521_pRDV_Eco_rev 5′-CTTTGAGAGGAGTCTTGAATTCGGA-3′ with an annealing temperature of 50 °C and 35 cycles using Vent polymerase (NEB). Respective bands were gel-purified using the Gel Purification Kit (Qiagen), and cDNA fragments were digested with BamHI and EcoRI followed by a clean-up, using the PCR purification kit (Qiagen) and ligated into pRDV. The resulting enriched pool served as template for the next round of selection using error-prone PCR and the primers T7B and TolAk.

In total, four rounds of selection were performed with increasing stringency lowering target protein concentration and increasing washing time. In rounds 3 and 4, an off-rate selection was implemented as described previously[48]. In round 3, error-prone PCR was applied to increase diversity, but no error-prone PCR was used in round 4, which instead served to efficiently enrich for generated high-affinity H2B binders. While rounds 1–3 were carried out with target immobilized on plates (see above), in round 4 the selection was performed in solution MyOne T1 streptavidin-coated magnetic beads (Thermo Fisher Scientific), as previously described[48].

Briefly, for round 3, a completion with a 1000-fold access of competitor was performed. For round 4, 20 nM of biotinylated ADP-ribosylated H2B peptide in the presence of 20 μM of ADP-ribosylated ARTD10cat (GST-fusion protein expressing the catalytic domain of ARTD10/PARP10 that is able to catalyze auto-mono-ADP-ribosylation) was used, and incubated for 1 h to increase the specificity toward the ADPr. A 30 min capture of Af1521 mutants that still were bound to the biotinylated peptide was performed using streptavidin-coated magnetic beads as previously described[48]. Af1521 mutants still binding to the biotinylated H2B peptide were eluted with EB buffer followed by RNA purification and RT-PCR.

The enriched pool of Af1521 mutants after the fourth round of ribosome display selection was subcloned via BamHI and PstI into pDST67[49] a pQE30

(QIAGEN) derivative, introducing an N-terminal MRGS(H)$_6$ tag. After transformation of *E. coli* XL-1 Blue, 92 single colonies were picked from selection agar plates, and expressed in a 96-well format in 2× YT media containing 100 μg/ml ampicillin and 1% glucose for 4 h at 37 °C after induction using 0.5 mM IPTG. Cells were harvested by centrifugation for 10 min at 4000 × $g$ and cells were lysed by addition of 50 μl B-PER II cell lysis buffer (Pierce), as previously described[48]. The lysate was diluted with 950 μl PBS-TB (PBS: 137 mM NaCl, 30 mM KCl, 80 mM Na$_2$HPO$_4$, 15 mM KH$_2$PO$_4$, pH 7.4 containing 0.1 (v/v) Tween20 and 0.2% (w/v) BSA), and cleared by centrifugation for 10 min at 4000 × $g$.

In order to identify initial hits, an ELISA was performed against the biotinylated and ADP-ribosylated H2B peptide and unmodified peptide as control. In brief, 96-well MaxiSorp plates (Nunc) were coated with 100 μl 66 nM Immunopure streptavidin (Pierce) in PBS overnight at 4 °C. After washing twice with PBS, the wells were blocked with PBS/0.5% BSA for 1 h at room temperature (RT). The biotinylated peptides were immobilized at a concentration of 100 nM in PBS-TB at RT for 1 h. After three washes with PBS-T, 100 μl of diluted peptide (90 μl PBS-TB plus 10 μl lysate) was added to the target-coated wells and incubated at RT for 1 h. Following three washes with PBS-T peptide-bound Af1521 mutants were detected using a mouse-anti-RGS(H)$_4$ primary antibody (1:5000, Qiagen, Cat. No. 34650) followed by a goat anti-mouse secondary antibody coupled to alkaline phosphatase (1:20,000, Sigma, Cat. No. A3562). After addition of the substrate (pNPP), absorbance at 405 nm was determined as previously described[48]. Ten clones were randomly selected for sequencing.

**Cloning and protein purification.** Randomly mutagenized Af1521 candidates were cloned into pGEX6P-1 (Addgene), using the restriction enzymes BamHI and EcoRI (NEB; pGEX6P-1_Af1521, forward primer 5′-GACAAAGGATCCATGGAAC GGCGTAC-3′, reverse primer 5′-CTTTGAGAGGAGTCTTGAATTCGGA-3′). Mutants of eAf1521 and WT Af1521 were obtained by site-directed mutagenesis (eAf1521-G42E, forward primer 5′-GAGCACGGCGAAGGGGTGGC-3′, reverse primer 5′-GCCACCCCTTCGCCGTGCTC-3′; eAf1521-R145K: forward primer 5′-GCT GGG ATA AAA GGC TGT GAT CTG-3′, reverse primer 5′-CAG ATC ACA GCC TTT TAT CCC AGC-3′; WT Af1521-K35E, forward primer 5′-GCC AACGAGAGGCTGG-3′, reverse primer 5′-CCAGCCTCTCGTTGGC-3′; eAf1521-E35E, forward primer 5′-GCCAACAAGAGGCTGG-3′, reverse primer 5′-CCAGCCTCTTGTTGGC-3′; eAf1521-I144G, forward primer 5′-CTGGGG GACGCGGC-3′, reverse primer 5′-GCCGCGTCCCCCAG-3′; WT Af1521-Y145R forward primer 5′-CTGGGATACGCGGCTGTG-3′, reverse primer 5′-CACAGC CGCGTATCCCAG-3′; eAf1521-R145Y, forward primer 5′-CTGGGATATACG GCTGTG-3′, reverse primer 5′-CACAGCCGTATATCCCAG-3′). His-tagged WT eAf1521 (with a N-terminal His tag) were constructed in the bacterial expression vector pET19b by GenScript (Piscataway, NJ, USA). For generation of Fc fusion constructs, fragments encoding the IL-2 secretion signal and an engineered mouse IgG2a Fc domain (described in ref. [59], Fc domain originally from pFUSE-mIg-G2ae1-Fc, InvivoGen) fused to an C-terminal HA and a His$_6$-tag, Af1521 WT and eAf1521 were sequence optimized and custom synthesized (GeneArt, Thermo Fisher Scientific), including restriction sites for further cloning. The fragment encoding the IL-2 secretion signal and the Fc tag was flanked by HindIII and XhoI restrictions sites, and KpnI and BamHI restriction sites were included in between the IL-2 secretion signal and the Fc tag sequences. Af1521 WT and eAf1521 fragments were each flanked by KpnI and BamHI restriction sites. The fragment containing the IL-2 secretion signal and the Fc domain with HA and His tags was cloned into pcDNA5/FRT/TO (Invitrogen), using the restriction enzymes HindIII and XhoI (NEB). The Af1521-encoding fragments were cloned in between the secretion signal and the Fc tag via KpnI and BamHI sites.

*Escherichia coli* BL21 was transformed with bacterial expression vectors, and protein expression was induced by adding 1 mM IPTG at OD$_{600}$ 0.4–0.6 for 3 h at 30 °C. Batch purification of GST-tagged or His-tagged proteins was carried out using glutathione Sepharose 4B beads (GE Healthcare) or ProBond™ Nickel-Chelating Resin (Thermo Fisher Scientific), according to the manufacturer's manual.

Fc fusion domains were expressed by transfecting HEK 293-T cells with the mammalian expression vectors using calcium phosphate. Roughly 6 h after transfection, the medium was removed and replaced with fresh DMEM containing 1% of FCS. Two days after transfection, the medium was collected and again replaced with fresh DMEM containing 1% of FCS. The collected supernatants were filtered through a 0.45 μm mesh, to eliminate residual cells. Batch purification of His-tagged Fc fusion domains from the supernatant was carried out using ProBond™ Nickel-Chelating Resin (Thermo Fisher Scientific), according to the manufacturer's manual. Expression and purification of all recombinant proteins were analyzed by sodium dodecyl sulfate–polyacrylamide gel electrophoresis (SDS–PAGE) followed by Coomassie staining or immunoblotting.

**Pull-down using ADP-ribosylated H2B peptide.** To test the binding of our engineered Af1521 candidates toward ADP-ribosylation, biotinylated ADP-ribosylated or non-modified H2B peptides[47] were bound to streptavidin Sepharose high-performance beads (GE Healthcare). For each pull-down, 5 μl of beads were washed three times in binding buffer (50 mM NaCl, 50 mM Tris-HCl pH 8, and 0.05% NP-40) and incubated overnight at 4 °C in 1 ml binding buffer with 2 μg of the modified or unmodified H2B peptide. Afterward, beads were washed three

times with 1 ml of incubation buffer (1× protease inhibitor cocktail (Roche), 50 mM Tris-HCl pH 8, and 0.05% NP-40) containing different salt concentration (50, 200, and 400 mM NaCl). A total of 2 μg of the recombinant protein was incubated with the beads in 1 ml of incubation buffer for 3 h at 4 °C. After centrifugation at 1500 × $g$ for 5 min, the supernatant containing unbound protein was added on 5 μl prewashed glutathione Sepharose 4B beads (GE Healthcare), and additionally incubated for 2 h at 4 °C. Subsequently, all beads were washed three times with incubation buffer before analysis by SDS–PAGE followed by Coomassie blue staining.

**Protein purification for crystallization.** A total of 11 mg of IMAC-purified N-terminally His6-tagged eAf1521 was further purified by SEC (Sephacryl-100; GE Healthcare) in 20 mM Tris pH 7.5, 300 mM NaCl, 10% glycerol, and 2 mM tris(2-carboxyethyl)phosphine (TCEP). The main peak fractions were concentrated by ultrafiltration in Vivaspin cartridges (Sartorius). Crystallization conditions were identified using the JCSG + crystal screen (Qiagen) and sitting drop vapor diffusion. Crystals grew at 4 °C in droplets consisting of 0.1 μl protein solution (23.1 mg/ml including 2 mM ADPr) and 0.2 μl of well solution (25% w/v PEG3350, 0.1 M Bis-Tris, pH 5.5, and either 0.2 M (NH$_4$)$_2$SO$_4$ or 0.2 M NaCl). Crystals were briefly transferred to cryo solution (well solution supplemented with 15% glycerol, 0.2 M NaCl, and 5 mM ADPr) and then stored under liquid nitrogen.

**Data collection, structure solution, and refinement.** Diffraction data were collected at the Diamond synchrotron radiation source, Didcot, UK, at beamline i24, at 0.96862 Å and a temperature of 277 K. We used AutoPROC[60] and XDS[61] for data processing and Phaser[62] for phasing. The structures were solved by molecular replacement using the WT Af1521 structure (PDB ID: 2BFQ[16]) as model template. Crystals grown in presence of (NH$_4$)$_2$SO4 (space group P6$_1$) yielded data down to 1.23 Å; however, the ADPr-binding site was involved in crystal contacts. To exclude artifacts, we abandoned refinement of this model. Crystals grown in presence of NaCl (space group C2) diffracted to 1.82 Å and the model was refined using Buster[63]. The progress of refinement was monitored using decreasing $R$ and $R_{\text{free}}$ values. The Ramachandran plot of the final model indicated 98.44% favored and 0.00% outliers. The Ramachandran plot was calculated using Molprobity[64]. Statistics from data collection and structure refinement are shown in Table 2.

**Surface plasmon resonance measurements.** SPR measurements were performed at 20 °C in HBS buffer (10 mM HEPES, 150 mM NaCl, 0.005% Tween-20, 25 μM EDTA, pH 7.6) using a Biacore T200 instrument (GE Healthcare). Proteins were immobilized on a (multi)NTA derivatized polycarboxylate hydrogel NiHC1000M chip, according to a protocol recommended by the chip manufacturer (Xantec, Düsseldorf, Germany). First, a 0.5 M EDTA solution (pH 8.5) was injected for 300 s, followed by a 120 s buffer injection, a surface activation step with 5 mM NiCl$_2$ for 60 s, and a 120 s injection of eAf1521 (280 nM) and 450 s WT Af1521 (215 nM), respectively, both containing a hexa-His tag. This was followed by a 120 s injection of immobilization buffer. All immobilization steps were performed in immobilization buffer (10 mM HEPES, 150 mM NaCl, 0.005% Tween-20, and 50 μM EDTA) at a flow rate of 5 μl/min. This results in a surface density of 390–520, and 4650 RU for eAf1521 and WT Af1521, respectively.

In the double-referenced binding experiments, twofold dilution series of five and seven concentrations of ADPr were injected for 30 s and 120 s for WT and eAf1521, respectively, at a flow rate of 30 μl/min in HBS buffer. ADPr concentrations were in the range of 2500–39 nM for binding to WT Af1521 and 25–1.25 nM for eAf1521, respectively. Sensorgrams measured for WT Af1521 at the ADPr concentrations measured for eAf1521 were not detectable, confirming that the engineering process indeed evolved a macro domain with higher affinity to ADPr. At higher ADPr concentrations, sensorgrams were deviating from 1:1 kinetics. For this reason, eAf1521 measurements were conducted with dilution series of only three concentrations (25–6.25 nM). In order to obtain a reliable dataset, three flow cells with immobilized eAf1521 at densities of 520, 500, and 390 RU, respectively, were used in parallel. In addition, WT Af1521 was measured under the conditions used for eAf1521. The chip surface was regenerated by injection of 0.5 M EDTA for 5 min. The chip was regenerated by a 5 min injection of 0.5 M EDTA in water.

The uncoated flow cell 1 of the sensor chip was used as a reference. Data were evaluated using Biacore software version 2.0.3. Sensorgrams were fitted using a 1:1 kinetic model. Sensorgrams of Af1521 were additionally evaluated using a steady-state (ss) model. The equilibrium dissociation constant of WT Af1521 calculated for ss conditions was four times larger, and an overestimation of the affinity in kinetic experiments is possible, presumably caused by rebinding as a common surface effect often observed in SPR measurements. The binding behavior of eAf1521 was governed by a 600 times slower dissociation and a two times faster association rate constant compared to WT Af1521. Calculation of the dissociation rate constant using $K_D$ (ss) resulted in a more reliable value (Table 1).

**In vitro ADP-ribosylation and de-ADP-ribosylation assays.** In vitro ADP-ribosylation assays were performed based on previously described methods[65,66]. For auto-ADP-ribosylation assays recombinant ARTD1 (10 pmol) was incubated in reaction buffer (RB; 50 mM Tris-HCl pH 7.4, 4 mM MgCl$_2$, and 250 μM

dithiothreitol (DTT)) with 100 μM NAD$^+$ and 200 nM of double-stranded annealed 40 bp long oligomer (5′-TGCGACAACGATGAGATTGCCACTACTTG AACCAGTGCGG-3′, 5′-CCGCACTGGTTCAAGTAGTGGCAATCTCATCGTT GTCGCA-3′) for 15 min at 37 °C. Recombinant ARTD8cat was incubated in RB buffer with either 100 μM NAD$^+$ or 200 nM [$^{32}$P] NAD$^+$ (Perkin Elmer) for 30 min at 37 °C. These reactions were stopped via the addition of SDS buffer or by filtering through an Illustra MicroSpin G-50 column (GE Healthcare), according to the manufacturer's protocol. De-ADP-ribosylation assays were performed in RB buffer. For de-modification of ARTD1, the auto-modified recombinant proteins were incubated with 10 pmol PARG for 30 min at 37 °C. The hydrolysis activities of WT Af1521 or eAf1521 were tested by incubating auto-modified ARTD8cat with either 10 pmol of recombinant WT Af1521, or eAf1521 for 2 h at 4 °C or 37 °C. De-modification of auto-modified ARTD8cat was then visualized by SDS–PAGE and autoradiography. For generation of PAR chains, poly-ADP-ribosylated ARTD1 was digested using proteinase K at 42 °C for 1 h.

**ADPr-modified peptide enrichment**. ADPr-modified peptide enrichments were carried out as previously described[26,67]. Untreated or H$_2$O$_2$-teated HeLa cells were washed twice with PBS, and subsequently lysed by adding at 95 °C preheated Gnd-HCl lysis buffer (6 M guanidine-hydrochloride, 5 mM TCEP, 10 mM chlor-oacetamide, in 100 mM Tris pH 8). After scraping and collecting the cells, the lysates were incubated at 95 °C for 10 min and sonicated for 1 min at an amplitude of 30%. After clearing the lysates by centrifugation at 2500 × $g$ for 5 min, the lysates were 10× diluted in 25 mM Tris pH 8, including 1 mM CaCl$_2$ and digested over-night using trypsin. The protease digestion was stopped by adding trifluoroacetic acid (TFA) to pH 4. After centrifugation at 4000 × $g$ for 10 min, the peptides in the supernatant were purified using reversed-phase Sep-Pak C18 cartridges (Waters). The peptides were eluted off the Sep-Pak using 50 and 80% acetonitrile. The acetonitrile was evaporated by vacuum centrifugation. The potential PARylated peptides were reduced to MARylated peptides using the enzyme PARG for 1 h at 37 °C in IP buffer (50 mM Tris-HCl, pH 8, 10 mM MgCl$_2$, 250 μM DTT, and 50 mM NaCl). The peptide mixtures were cooled down before incubation with glutathione Sepharose 4B beads (GE Healthcare) coupled either to WT Af1521 or to eAf1521 for 2 h at 4 °C. Afterward, beads were washed three times with ice-cold IP buffer, once with ice-cold water followed by three elution steps with 100 μl 0.15% TFA. Eluted peptides were desalted using reverse-phase C18 StageTips.

**Liquid chromatography and mass spectrometry analysis**. Identification of ADP-ribosylated peptides from untreated and H$_2$O$_2$-treated HeLa cells was per-formed on an Orbitrap Fusion Tribrid mass spectrometer (Thermo Fisher Scien-tific), coupled to an ACQUITY M class UPLC liquid chromatograph (Waters). We applied an ADPr product-dependent analysis called HCD-PP-EThcD[54]. Solvent compositions in channels A and B were 0.1% formic acid in water, and 0.1% formic acid in acetonitrile, respectively. Peptides were loaded onto a nanoEase M/Z Symmetry (Waters) trap column, 180 μm × 20 mm, packed with C18 material, 5 μm, 100 Å, and separated on an analytical nanoEase M/Z HSS T3 Column (Waters, 75 μm × 200 mm) packed with reverse-phase C18 material (Waters, 1.8 μm, 100 Å). Peptides were eluted over 110 min at a flow rate of 300 nl/min. A linear elution gradient protocol from 3 to 25% B for 95 min, followed by 35% B for 5 min, and a wash step at 95% B for 5 min, respectively, was used. Full-scan MS spectra (350−2000 $m/z$) were acquired at resolution of 120,000, with AGC target set at 4e5 and a maximum injection time of 50 ms. High-energy HCD MS/MS spectra (collision energy at 38%) were acquired at a resolution of 30,000 with the AGC target set at 5e4 and a maximum injection time of 60 ms. Two or more observed ADPr fragment peaks (136.0623, 250.0940, 348.07091, and 428.0372) in the high-energy data-dependent HCD scan triggered additional high-quality HCD and EThcD MS/MS scans (resolution of 120,000, AGC target set at 5e5, injection time of 240 ms, collision energy for HCD scan at 35%).

**MS data analysis and label-free quantification**. MS1-based LFQ was performed by applying Progenesis QI for Proteomics software (v. 3.0.6039.34628, Nonlinear Dynamics, Purham, NC) with default settings and the following exceptions. Pep-tide ions were filtered for charges ranging from +2 to +5. A maximum of the five top-ranked MS/MS spectra per peptide ion were exported with the most intense 200 peaks per spectrum with activated charge-deconvolution and deisotoping option as a Mascot generic formatted file (MGF). MS/MS spectra were searched with Mascot for each type of fragmentation (HCD and EThcD). Mascot searches were carried out as previously described[67]. A peptide tolerance of 8 ppm and MS/MS tolerance of 0.03 Da were used. Enzyme specificity was set to trypsin allowing up to four missed cleavages. The MGFs were searched against the target-decoy UniProtKB human database (taxonomy 9606, canonical sequences and reviewed entries only, downloaded on 2019/07/06). N-terminal protein acetylation was set as a variable modification. S, R, K, D, E, and Y residues were set as variable ADPr acceptor sites with a mass shift of 541.0611 Da. Carbamidomethylation was set as a fixed modification on C and oxidation as a variable modification on M. The neutral losses from the ADPr 249.0862, 347.0631, and 583.0829 Da were scored in HCD fragment ion spectra. For HCD and EThcD fragment ion spectra, the marker ions at $m/z$ 428.0372, 348.0709, 250.0940, and 136.0623 were ignored for scoring. The Mascot search results were imported into Scaffold and filtered for protein and

peptide FDR values of 2% and 1%, respectively. When multiple precursors were observed for the same peptide, the values were summed up to obtain the total intensity level of the peptide.

For ADPr-modified peptide and protein analysis, MS and MS/MS spectra were converted to MGF using Proteome Discoverer, v2.1 (Thermo Fisher Scientific, Bremen, Germany). Separate MGF files were created from the raw file for each type of fragmentation (HCD and EThcD), using a dedicated rule in the converter control[68]. Mascot was used as described above and the Mascot search results were imported into Scaffold 4 software (version 4.8.4). Peptides were considered as correctly identified when a Mascot score >15 and a Mascot delta score >5 were obtained. These settings ensured a FDR <1% at the PSM level. For PTM localization site, probability estimation ScaffoldPTM software (version 3.2.0) was used invoking the site localization algorithm Ascore[69], including neutral losses for HCD fragment ion spectra. Razor peptides, spectra that belong to more than one peptide, were not included for further analysis on the peptide and protein level. Proteins that contained the same peptides and could not be differentiated based on MS/MS analysis were only reported once for further analysis on the protein level. For the ADP-ribose amino acid acceptor site analyses, peptides identified with EThcD fragmentation, having a localization score >95%, were used if not stated otherwise.

**Bioinformatic analyses**. Statistical analysis, scatter plot, bar plot, and volcano plot analysis were performed using Prism 8. Normalized LFQ intensities were imported. For statistical analysis, the log$_{10}$-transformed and normalized MS1 signal intensity of three biochemical replicates were compared using a Student's $t$ test, two-sided. Venn diagrams were generated using BioVenn (http://www.biovenn.nl).

**Pull-down using PARylated proteins**. To test the binding of our eAf1521 toward PARylation, recombinant His-tagged ARTD1 was in vitro poly-ADP-ribosylated as described above. A total of 25 pmol of auto-modified ARTD1 was incubated with 125 pmol of either WT Af1521, eAf1521, or GST in 1 ml binding buffer (1% BSA, 50 mM NaCl, 50 mM Tris-HCl pH 8, and 0.05% NP-40) at 4 °C for 1 h. A total of 10 μl prewashed glutathione Sepharose 4B beads (GE Healthcare) were added and additionally incubated for 1 h at 4 °C. After centrifugation at 1500 × $g$ for 5 min, the supernatant containing unbound PARylated ARTD1 was added on 10 μl pre-washed ProBond™ Nickel-Chelating Resin (Thermo Fisher Scientific), and addi-tionally incubated for 2 h at 4 °C. Subsequently, all beads were washed three times with binding buffer, and one additional time with binding buffer lacking BSA before analysis by immunoblotting.

**Immunoblotting**. Untreated or treated HeLa cells were lysed with RIPA buffer (50 mM Tris-HCl pH 7.4, 400 mM NaCl, 1% NP-40, 0.1% Na-deoxycholate, 1× protease inhibitor cocktail (Roche), and 10 μM PJ34), sonicated, and centrifuged at 16,000 × $g$ for 10 min.

HeLa lysates or recombinant proteins were mixed with SDS buffer, and boiled at 95 °C for 5 min. After separation by SDS–PAGE, a wet transfer onto PVDF membrane was performed. The membranes were blocked with 5% milk in TBS-T for 1 h at RT. Primary antibodies were diluted in 5% milk in TBS-T and incubated at 4 °C overnight. After three washes with TBS-T for 5 min, the secondary antibody (in TBS-T) was incubated for 1 h at RT and the membranes were additionally washed three times with TBS-T. For dot blot analysis, proteins were vacuum blotted onto a nitrocellulose membrane that was further blocked in milk and stained with antibodies as described above.

The bands or dots were visualized using the Odyssey infrared imaging system (LICOR). The following primary and secondary antibodies were used for immunoblot and dot blot analyses: anti-tetra-His (1:1000, Qiagen, Cat. No. 34670, Lot No. 160043206), anti-GAPDH (1:1000, Santa Cruz Biotechnology, Cat. No. sc-25778, Lot No. B0106) Fc-WT Af1521 (1:400, 500 ng/ml, this paper), Fc-eAf1521 (1:400, 500 ng/ml), 10H (PAR antibody, 1:1000, made in-house), IRDye 800CW goat anti-rabbit IgG (1:15,000, LI-COR, P/N 925-32211), and IRDye 680RD goat anti-mouse IgG (1:15,000, LI-COR, P/N 925-68070). Molecular weights are indicated by the PageRuler Plus Prestained Protein Ladder (Thermo Fisher Scientific).

**Immunofluorescence**. For IF experiments, HeLa cells were grown on glass cov-erslips. After treatment, HeLa cells were fixed with 4% PFA for 15 min at RT, and permeabilized for 10 min at RT in PBS supplemented with 0.2% Triton X-100 (Sigma Aldrich). After blocking the cells with PBS supplemented with 10% of goat serum for 1 h, the cells were incubated with the primary antibody (diluted in blocking solution) overnight at 4 °C. The cells were washed two times with PBS and subsequently incubated with the secondary antibody (diluted in blocking solution) for 2 h at RT. After two 5 min washes with PBS, the cells were incubated with 0.1 μg/ml DAPI in PBS for 20 min at RT. The cells were additionally washed twice with PBS for 5 min. Then the coverslips were briefly washed in distilled water and mounted on glass slides, using 5.5 μl Mowiol solution per coverslip. The following primary and secondary antibodies were used for IF analyses: Fc-WT Af1521 (1:400, 500 ng/ml, this paper), Fc-eAf1521 (1:400, 500 ng/ml, this paper), and goat anti-mouse secondary antibody (1:500, Thermo Fisher Scientific, Cat. No. A-11029). For all images, brightness and contrast were adjusted using FIJI. For all images

within one experiment, the same acquisition and image processing settings were used.

**Reporting summary**. Further information on research design is available in the Nature Research Reporting Summary linked to this article.

## Data availability

The MS proteomics data generated and analyzed during the current study have been deposited to the ProteomeXchange Consortium via the PRIDE[70] partner repository with the dataset identifier PXD016686, and are available from the corresponding author on reasonable request. MS proteomic data are displayed in the following figures: Fig. 2a–f, Supplementary Fig. 2b–g, Fig. 3a–c, Supplementary Fig. 3a–c and Supplementary Data 1–4. Coordinates and structure factors from eAf1521 generated and analyzed during the current study have been deposited to the Protein Data Bank (PDB) under PDB ID 6FX7. Coordinates and structure factors from WT Af1521 have been obtained from the PDB (PDB ID 2BFQ). Structural data are displayed in the following figures: Fig. 1c–d and Supplementary Fig. 1h. Source data are provided with this paper.

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

## Acknowledgements
We thank Tobias Suter (University of Zurich) for the helpful discussions and for providing editorial assistance. We thank the Center for Microscopy and Image Analysis (ZMB) and the Functional Genomics Center of the University of Zurich (FGCZ) for services and assistance. We especially thank Peter Gehrig, Paolo Nanni, Christian Panse, and Bernd Roschitzki from the FGCZ for helpful suggestions and discussions, as well as bioinformatic support. We thank Alexandra Golzmann (RWTH Aachen University) for technical support. We thank Gerhard Müller-Newen (RWTH Aachen University) for Fc plasmids. B.L. is supported by the Deutsche Forschungsgemeinschaft LU 466/16-2 and H.S. by the Swedish Cancer Society (2017-492). ADP-ribosylation research in the laboratory of M.O.H. is funded by the Kanton of Zurich and the Swiss National Science Foundation (grant 31003A_176177).

## Author contributions
Project conceptualization and administration: M.O.H. and K.N. (lead), and F.R., D.M.L.P, and A.P. (supporting). Investigation: K.N. (lead), F.R., B.D., M.B., T.K., J.G., J.S., A.-G.T., D.M.L.P., and R.I. (supporting). Methodology: A.P. and B.D. (lead for ribosome display), and F.R. and R.I. (supporting); H.S. (lead for crystal structure and modeling) and T.K. and A.-G.T. (supporting); J.S. (lead for SPR) and K.N. (supporting). K.N. (lead MS), and J.G., D.M.L.P., and (supporting); M.B. and B.L. (lead for Fc fusion), and K.N. (supporting). Data curation and formal analysis: K.N. (lead); J.G. (supporting for MS analysis); H.S. (lead for crystal structure); J.S. (lead for SPR). Visualization and validation: K.N. (lead), and T.K. and H.S. (supporting). Writing, review, and editing of MS: M.O.H., K.N., and D.M.L.P. (lead), A.P., B.D., and H.S. (supporting), and F.R., M.B., T.K., J.G., J.S., A.-G.T., and B.L. (editing).

## Competing interests
The authors declare no competing interest.
