## [Peer Review File · Nature Communications]

Reviewers' comments:

Reviewer #1 (Remarks to the Author):

ADP-ribosylation is a protein posttranslational modification that is mediated by a class of enzymes called PARPs. Although protein ADP-ribosylation is known to regulate many aspects of cell stress responses, its exact signaling mechanisms are still poorly understood. In this manuscript, Nowak et al., performed random mutagenesis and identified a novel protein domain (eAf1521) that binds to ADP-ribose. Compared to the previous archaeal Af1521 macro domain, eAf1521 displays ~1000-fold higher affinity towards ADP-ribose. The authors determined the crystal structure of the engineered Af1521 domain, and showed that the enhanced affinity could be explained by the trapping of ADP-ribose induced by two key amino acid substitutions. eAf1521 was then deployed to show that it leads to better enrichment of the ADP-ribosylated peptides, and their proteomic identification. Furthermore, the authors showed the utility of eAf1521 as an ADP-ribose detection reagent in immunoblotting and immunofluorescence experiments. The manuscript was well-written, and this novel ADP-ribose binding domain could be useful to many researchers in the PARP field and beyond.

Specific comments:

- (1) Line 428, what is the binding affinity of eAf1521 towards Oligo- and Poly-ADP-ribose? Does the binding differ between free ADP-ribose and ADP-ribose that is attached to a protein substrate?
- (2) Line 438, have the authors tested the eAf1521-induced hydrolysis of ADP-ribosylated proteins using longer treatment times?
- (3) Line 476, what is the enrichment efficiency of eAf1521 towards PARylated proteins? The authors need to incubate whole cell lysates with eAf1521, and then measure the level of PAR signal in input and flow-through fractions.
- (4) Line 477, what are the elution conditions? Because eAf1521 binds very tightly to ADP-ribose ($K_d = 3 \text{ nM}$), the authors need to show that all ADP-ribosylated peptides could be released (i.e., no bias is introduced at the elution step).
- (5) Line 606, what is the source of the extra-nuclear ADP-ribose signal? Is this mono- or poly-ADP-ribose? Can the authors speculate which PARP(s) is involved?

Reviewer #2 (Remarks to the Author):

The manuscript authored by Kathrin Nowak et al. and entitled “Engineering Af1521 improves ADP-ribose binding and identification of ADP-ribosylated proteins” reports the engineered archaeal Af1521 macro domain with 1000-fold increased ADPr-binding affinity, and provides substantial experimental evidence of the engineered protein as an immuno-tool for the improved detection of ADP-ribosylation. This paper is well-written and most of the conclusions in the paper are supported by the results.

The major comments are listed below.

1. Line 639: Although SPR measurement could estimate the binding affinities already, ITC can provide fundamental thermodynamic parameters that can be used to establish the physiochemical origin of molecular interactions (entropy-driven or enthalpy-driven, H-bond or hydrophobic interaction favor). The application of ITC for this study and discussion based on the data are suggested.
2. Line 457: “The novel conformation of eAf1521 decreases polypeptide flexibility while strengthening the interaction between the protein and the terminal ribose”. The evidence of “flexibility decrease” should be better provided by MD simulation or B-factor comparison.
3. Line 459: “Second, formation of a prominent tunnel enclosing the ADPr pyrophosphate was observed and augments binding”. But the tunnel formed by a closed loop should not allow entry of ligand. In other words, decreasing flexibility of loop make binding pocket (tunnel) rigid, which against the explanation above.
4. Since the orientation of the terminal ribose moiety in ADPr bound to eAf1521 was different from that bound to WT, figure including electron density map and the ligand should be provided for the convincement.
5. Dose such orientation of the terminal ribose of ADPr make C2” position buried in the pocket? Or it would change the substrate (binding sequence) preference, not only binding affinity?
6. Line 666: If Af1521 alters its favor ADPr acceptor sites (binding sequence) after engineering, from MS data, is there any peptide bound to WT Af1521, but not eAf1521?

Reviewer #3 (Remarks to the Author):

Nowak and colleagues have successfully engineered a variant of an archaeal ADP-ribose (ADPr) binding protein (Af1521) that displays greater-than 1000-fold binding affinity for ADPr modified protein substrates over the wild-type protein. This construct represents a significant step forward in the development of ADPr-specific enrichment tools and will aid in the characterization of the ADPr-modified proteome, the visualization of ADPr containing proteins via Western blot analysis, and improve the detection of ADPr in vivo using immunofluorescence. Importantly, the development of a

tight ADPr binding protein will allow for the identification of novel protein substrates that have thus far evaded detection. This is a very useful and interesting article that will boost various proteomic and biochemical research efforts in the field. The following are a few minor specific comments regarding the manuscript.

1. In the proteomic analysis, it was noted that variable ADPr acceptor sites were allowed to occur on S, R, K, D, E, and Y. Was there a reason that C was not included in this list? Had the authors based this on previous reports or was it not observed in their preliminary work flow? It may be worth addressing why C was not included.

2. On line 429, the authors claim that Af1521 "displayed a KD > 1000 times higher than WT Af1521." Should this read lower instead?

3. The analysis of the Y145R mutation is very clear and is supported by the loss of function alteration from Arg to Lys. In this vein, it might be worth demonstrating that the mutation of I141 to either Leu or Val causes the loss of binding affinity for Af1521 and that the proposed phosphate trapping contributes significantly towards the increase in overall affinity.

4. In the discussion of the overlap between Af1521 and eAf1521-dependent H₂O₂-induced proteomes, the authors mention that they observe a 95% overlap between the proteins identified using WT Af1521 and the larger eAf1521 dataset. In the remaining 5% of proteins lost from WT Af1521 to eAf1521, is there any pattern to the lost proteins (i.e. a biological, structural, or sequence-based reason for why eAf1521 didn't capture these proteins)?

5. The authors claim that the selection strategy that produced eAf1521 did not result in any apparent bias in the selection of ADPr-modified proteins. It might be useful to include a comparison of their current method with previously published proteomic efforts (with antibody, Af1521, or PARG based enrichment) using H₂O₂-induction to highlight the overlap of the current method with the literature. This would help strengthen their claim regarding the lack of bias.

6. When the authors attempted their method using decreasing amounts of HeLa lysate, they note that the lowest sample (5 mg) was unable to detect ~20% of the proteins identified at the 20 or 10 mg level. Of the 20% that was lost, was this primarily due to the lower abundance of those proteins in the HeLa cells?

7. In Fig 4A, the authors present the Western blots comparing WT-Af1521 and eAf1521 based detection. Were these blots taken at the same time with the same exposure? I assumed that they were from the same raw image, but it would be worth explicitly stating this in the figure legend.

Ian Carter-O'Connell

Reviewer #4 (Remarks to the Author):

Nowak et al describe the engineering of a new variant of Af1521, a naturally occurring protein that binds ADP-ribosylated proteins, using error-prone PCR, ribosome display, and expression of recombinant protein in *E. coli*. The affinity of the engineered variant for ADP-ribosylated proteins is 1,000x higher than that of the wild-type Af1521, which makes it a more sensitive detection tool for ADP-ribosylated proteins, e.g., in proteomic applications. The comparison between crystal structures of the wild-type and the variant suggests that the increased affinity is due to a change of orientation of ADP-ribose in the binding site of Af1521.

The strength of this manuscript is in the magnitude of the improvement of the protein being engineered, in the high-resolution crystal structure that allows the interpretation of this improvement at atomic level, and in the demonstration of improved utility of the variant as a detection tool.

The major weaknesses of the manuscript are the gaps in the description of the materials and methods used during ribosome display and screening of the selected variants, the poorly edited text, and an occasional leap to conclusion without supporting evidence. Also, I wish they had used site-directed mutagenesis of the engineered variant to determine which of the 9 mutations in the variant are essential for the improved binding phenotype, and which are incidental.

Specific comments follow:

- Line 62: "Several different proteins able to bind poly-ADPr (e.g. PAR binding motif (PBM)), two consecutive ADPr units (e.g. PAR-binding zinc finger (PBZ)) or iso-ADPr (e.g. WWE domain) have been described (i.e. readers)." Missing reference. Also, it's not clear what "readers" refers to.

- Line 77: “our understanding of the cellular processes regulated by ADP-ribosylation had remained very modest for a long time due to the limited tools available (e.g. antibodies).” What traditional tools were “limited,” and in what way were they “limited”? Antibodies have done very well in other applications. How have they failed in this field?
- Line 110: “The randomization occurs either at a low rate by the intrinsic error rate of the polymerase used but can be enhanced using error-prone PCR amplification methods (51), DNA shuffling (52), or both.” There are additional and very common sequence-diversification methods, most of which take advantage of in vitro oligonucleotide synthesis, using mixtures of nucleotides or of codons. Reference (52) is 25 years old.
- Line 139: The lack of detail about how the ribosome-display selection was performed is puzzling. For example: “After error-prone PCR, the mutagenized Af1521 constructs were transcribed and translated in vitro as described earlier (53). For the first selection round, 200 nM of biotinylated ADP-ribosylated H2B peptide (Fig. S1) was immobilized on a precoated streptavidin plate. The ribosome complexes displaying in vitro translated mutagenized Af1521 domains were added to the wells and incubated for 1 hr at 4 °C, and subsequently each well was immediately washed twice and additionally 4 times for 2 min. Afterwards a reverse transcription was performed using the recovered mRNA and subsequently amplified by PCR using the scaffold-specific primers Af1521_pRDV_fwd_2 5'-gacaaaggatccATGGAACGGCGTAC-3' and Af1521_pRDV_Eco_rev 5'-CTTTGAGAGGAGTCTTgaattcgga-3'.” The sequences of the PCR primers are given, but very little is revealed about the selection protocol. What was the volume at each round? What was the source of the streptavidin plates? The amount of RNA/DNA? The buffer? The authors state that transcription and translation followed the protocol in (53), but don't tell us anything about the selection protocol. If they did follow the selection protocol in reference (53) exactly, they need to state that. If they followed it mostly but made some changes, the changes need to be specified.
- Line 157: “As competitor 20 nM of automodified ARTD10cat was used to increase the specificity towards the ADP-ribosylated H2B peptide.” What is ARTD10cat? What was the competition protocol? How did that increase the specificity?
- Line 161: “After 96 well expression of the mutagenized Af1521 domains and cell lysis using B-PERII (Pierce) as previously described (45), an ELISA was performed against the ADP-ribosylated H2B peptide (biotinylated and immobilized on streptavidin plates) using 10 µL of bacterial lysate. The binding was detected using the MRGS-H6-tag (45).” How did the variants enriched during the selection make it from line 160 to line 161? How many wells or 96-well plates of variants were expressed? What reagents and conditions were used in the ELISA? At what stage were the clones sequenced? How many different variants of Af1521 were purified and characterized? How did the final lead compare to the other variants considered?
- Line 411: “Many clones showed a significant binding signal to the ADP-ribosylated H2B peptides over the signal of the unmodified peptide (data not shown).” This data needs to be shown, at least in supplementary materials. How many clones were tested? How many bound? How is “significant” binding defined?
- Lines 464-472 describe a limited attempt to explore the contribution of a salt bridge between two mutated residues to the increase in affinity for ADPr. Only a single mutation from R to K was made,

and the resulting reduction in affinity led the authors to attribute the high affinity of the engineered variant to the removed guanidine group. In the absence of the crystal structure of this RK mutant, we don't know what effects on the structure and function this mutation had. A more extensive mutagenesis experiment may be able to pinpoint the role of this salt bridge, and the roles of the other 7 mutations, in improving the affinity for ADP-ribosylated proteins.

- Line 672: The authors argue that the engineered detection reagent is unlikely to show a bias for the peptide that it was selected against given that it has pulled down unrelated proteins also. Then they follow with, "Nevertheless, to overcome any potential bias that eAf1521 may exhibit towards certain ADPr acceptor sites, we would recommend performing the enrichment of ADP-ribosylated peptides with both WT Af1521 and eAf1521 in parallel or in tandem." This recommendation is surprising given that the work described was directed towards identifying a more sensitive and thus more useful detection reagent, and given that this goal appears to have been achieved. Instead of continuing to rely on the older and less sensitive reagent, let's test both reagents against a range of different ADP-ribosylated proteins and compare their affinities, in vitro, to determine whether there is such a bias.

- The manuscript is difficult to read and would benefit from careful editing for grammar and style.

Response to Reviewer's Comments on Manuscript

We thank the reviewers for evaluating our manuscript and for the constructive criticism. The reviewer suggestions helped to improve the quality of the manuscript.

Below, we have outlined the additional new experiments and specific changes made to the manuscript. Changes in the manuscript have been highlighted in yellow. Minor changes (like typos, grammatical errors and slight changes in wording and panels) were not highlighted.

Furthermore, we provide a detailed point-by-point response to each of the reviewer's comments. Overall, the additional experiments confirmed our previous conclusions and support the statement that eAf1521 improves ADP-ribose binding and identification of ADP-ribosylated proteins, thus allowing the detection of novel ADP-ribosylation events.

Summary of changes made to the figures and figure legends:

Main Figures	Suppl. Figures
Fig. 1A	Suppl. Fig. 1A (before 1B)
Fig. 1B	Suppl. Fig. 1B (before 1A)
Fig. 1C	Suppl. Fig. 1C, new
Fig. 1D	Suppl. Fig. 1D, new
Fig. 1E, new	Suppl. Fig. 1E, new
	Suppl. Fig. 1F (before 1C)
Fig. 2A	Suppl. Fig. 1G (before 1D)
Fig. 2B	Suppl. Fig. 1H, new
Fig. 2C	Suppl. Fig. 1I, (before 1E)
Fig. 2D	
Fig. 2E	Suppl. Fig. 2A
Fig. 2F	Suppl. Fig. 2B
	Suppl. Fig. 2C
Fig. 3A	Suppl. Fig. 2D
Fig. 3B	Suppl. Fig. 2E
	Suppl. Fig. 2F, new
Fig. 4A	Suppl. Fig. 2G, (before 2F)
Fig. 4B	
	Suppl. Fig. 3A
	Suppl. Fig. 3B
	Suppl. Fig. 3C
	Suppl. Fig. 4A
	Suppl. Fig. 4B
	Suppl. Fig. 4C
	Suppl. Fig. 4D
	Suppl. Fig. 4E, new
	Suppl. Fig. 4F, new
	Suppl. Fig. 4G, new
	Suppl. Fig. 5A
	Suppl. Fig. 5B
	Suppl. Fig. 5C

Point-by-point response to the questions raised by the reviewer:

Reviewer #1 (Remarks to the Author):

ADP-ribosylation is a protein posttranslational modification that is mediated by a class of enzymes called PARPs. Although protein ADP-ribosylation is known to regulate many aspects of cell stress responses, its exact signaling mechanisms are still poorly understood. In this manuscript, Nowak et al., performed random mutagenesis and identified a novel protein domain (eAf1521) that binds to ADP-ribose. Compared to the previous archaeal Af1521 macro domain, eAf1521 displays ~1000-fold higher affinity towards ADP-ribose. The authors determined the crystal structure of the engineered Af1521 domain, and showed that the enhanced affinity could be explained by the trapping of ADP-ribose induced by two key amino acid substitutions. eAf1521 was then deployed to show that it leads to better enrichment of the ADP-ribosylated peptides, and their proteomic identification. Furthermore, the authors showed the utility of eAf1521 as an ADP-ribose detection reagent in immunoblotting and immunofluorescence experiments. The manuscript was well-written, and this novel ADP-ribose binding domain could be useful to many researchers in the PARP field and beyond.

Response: We thank this reviewer for the overall positive evaluation of our work.

Specific comments:

(1) Line 428, what is the binding affinity of eAf1521 towards Oligo- and Poly-ADP-ribose? Does the binding differ between free ADP-ribose and ADP-ribose that is attached to a protein substrate?

Response: To test whether eAf1521 is able to bind Oligo- and Poly-ADP-ribose (ADPr), we performed in vitro ADP-ribosylation assays using recombinant ARTD1 modified with low and high NAD⁺ concentrations. Immunoblot analysis using Fc-WT Af1521 or Fc-eAf1521, confirmed that eAf1521 strongly binds both Oligo- and Poly-ADP-ribosylated ARTD1 (new Suppl. Fig. 4E, page 16). In addition, we now provide evidence that eAf1521 recognizes also isolated PAR chains that are not attached to a protein substrate (new Suppl. Fig. 4G, page 16 and 17).

To further investigate binding of the eAf1521 to free ADP-ribose and ADP-ribose attached to a target, we performed competition experiments using the ADP-ribosylated H2B peptide (new Suppl. Fig. 1E, page 12). Increasing ADP-ribose concentrations strongly affected the binding capacity of WT Af1521 towards the ADP-ribosylated H2B peptide in a concentration dependent manner. In contrast, eAf1521 was able to bind the ADPr target even in presence of 10x higher concentration of free ADP-ribose. In addition, competition experiments for immunoblotting and immunofluorescence using Fc-eAf1521 in presence of 100x higher ADPr concentration did not completely abolish the signal detected with eAf1521 (Suppl. Fig. 4D and Suppl. Fig. 5E). Together, these experiments provide additional evidence that eAf1521 is binding stronger to ADPr, independent whether the moiety is peptide bound or in the context of Oligo- or Poly-ADP-ribose.

(2) Line 438, have the authors tested the eAf1521-induced hydrolysis of ADP-ribosylated proteins using longer treatment times?

Response: In general, de-ADP-ribosylation reactions are performed with shorter incubation times from 15 min up to 1 hr (REFs: Abplanalp et al., Fontana et al., Rosenthal et al.). Here, our hydrolysis reactions were incubated at the indicated temperatures for 2 hrs and resulted in complete demodification of ARTD8cat by Af1521 and eAf1521 under physiological conditions (37°C) (Suppl. Fig. 1 G). Moreover, the aim of the hydrolysis experiment was to provide evidence that under the conditions used for the MS-based ADP-ribosylome workflow (4°C, 2 hrs) demodification of ADP-ribosylated proteins was not observed. Furthermore, the mutations of eAf1521 did not alter this hydrolase activity compared to WT Af1521. Based on these results, we have not test additional time points but have revised the wording in the manuscript to clarify this (page 13).

(3) Line 476, what is the enrichment efficiency of eAf1521 towards PARylated proteins? The authors need to incubate whole cell lysates with eAf1521, and then measure the level of PAR signal in input and flow-through fractions.

Response: As requested by this reviewer we have performed an additional pull-down assay to investigate the enrichment of PARylated proteins (new Suppl. Fig. 4F). Instead of using whole cell lysate, we decided to use recombinant poly-ADP-ribosylated ARTD1 as PAR substrate to have a more standardized PARylation input sample in which there are no potential erasers that could alter the PARylation state of the samples during the

incubation time. The enrichment revealed, that GST-WT Af1521 and GST-eAf1521 are both capable to enrich PARylated ARTD1 compared to the GST only control (page 16).

(4) Line 477, what are the elution conditions? Because eAf1521 binds very tightly to ADP-ribose ($K_d = 3$ nM), the authors need to show that all ADP-ribosylated peptides could be released (i.e., no bias is introduced at the elution step).

Response: The bound peptides were released by adding SDS-loading buffer and heat denaturing the samples. We are, thus, sure that all ADPr peptides were released and no bias was introduced in the pulldown experiments. We have now included this additional experimental evidence demonstrating that the elution was efficient and that the amount of peptides was equal among the different binding conditions (new Suppl. Fig. 11).

(5) Line 606, what is the source of the extra-nuclear ADP-ribose signal? Is this mono- or poly-ADP-ribose? Can the authors speculate which PARP(s) is involved?

Response: As requested by this reviewer, we have included a statement regarding the source and the nature of the ADP-ribosylation in our revised discussion. Since the extranuclear signal is not detected using a conventional anti-PAR antibody (1), the nature of the extranuclear signal is rather mono-/oligo-ADP-ribosylation. The exact cellular localization of this signal as well as its potential function and importance is out of the scope of this study but is currently under investigation (page 19).

Reviewer #2 (Remarks to the Author):

The manuscript authored by Kathrin Nowak et al. and entitled “Engineering Af1521 improves ADP-ribose binding and identification of ADP-ribosylated proteins” reports the engineered archaeal Af1521 macro domain with 1000-fold increased ADPr-binding affinity, and provides substantial experimental evidence of the engineered protein as an immuno-tool for the improved detection of ADP-ribosylation. This paper is well-written and most of the conclusions in the paper are supported by the results.

Response: We thank this reviewer for proof-reading our manuscript and constructive feedback.

The major comments are listed below.

1. Line 639: Although SPR measurement could estimate the binding affinities already, ITC can provide fundamental thermodynamic parameters that can be used to establish the physiochemical origin of molecular interactions (entropy-driven or enthalpy-driven, H-bond or hydrophobic interaction favor). The application of ITC for this study and discussion based on the data are suggested.

Response: The crystal structure of the eAf1521 macrodomain provides strong evidence that ADP-ribose localizes to the ADP-ribose binding cleft in a similar manner as with the wild type protein due to the virtually identical overall structure (Fig. 1C). Therefore, for the purpose of this study, we were not primarily interested in the thermodynamic properties of ADPr binding. More critical for us was to determine the binding affinity and the kinetics of binding. Therefore, SPR was the method of choice.

2. Line 457: “The novel conformation of eAf1521 decreases polypeptide flexibility while strengthening the interaction between the protein and the terminal ribose”. The evidence of “flexibility decrease” should be better provided by MD simulation or B-factor comparison.

Response: We thank the referee to point this out. We do not have any evidence supporting a significant decrease in flexibility in the peptide chain. In the light of this, we have eliminated this statement and the sentence now reads: Indeed, the novel eAf1521 amino acid arrangement appears to contribute directly to strengthening the interaction to the proximal ribose. (revised on page 13).

3. Line 459: “Second, formation of a prominent tunnel enclosing the ADPr pyrophosphate was observed and augments binding”. But the tunnel formed by a closed loop should not allow entry of ligand. In other words, decreasing flexibility of loop make binding pocket (tunnel) rigid, which against the explanation above.

Response: The tunnel is formed as a consequence of ADP-ribose binding by the Ile144 side chain that is in a very similar position in the wild type protein. However, for eAf1521, where the terminal ribose flips by hydrogen bonding to Arg145, the ribose ring is now accessible and induces a rotation of the Ile144 side chain to reside over the pyrophosphate moiety. Thus, presence of the terminal ribose in the conformation that is observed in eAf1521 is a prerequisite for the formation of this tunnel, as stated in the last sentence of the paragraph. Our

crystallographic observations are not in conflict with the results of our SPR experiments. We have revised this section for clarity: The hydrogen bonding interactions between R145 of eAf1521 and the bound terminal ribose also changes the rotamer of the I144 side chain (Fig. 1D, revised page 13).

4. Since the orientation of the terminal ribose moiety in ADPr bound to eAf1521 was different from that bound to WT, figure including electron density map and the ligand should be provided for the conviction.

Response: We have created a new figure showing the electron density around the bound ADP-ribose, the R145-E35 side chains, as well as the side chain of N34 that interacts with ADPr (new Suppl. Fig. 1E).

5. Does such orientation of the terminal ribose of ADPr make C2'' position buried in the pocket? Or it would change the substrate (binding sequence) preference, not only binding affinity?

Response: The C2'' oxygen is not buried but it engages in a hydrogen bonding interaction with the side chain of N34 at the surface of the domain. From the crystal structure it is impossible to predict the consequence of this on binding selectivity and affinity. However, it is clear that both the C1'' and C2'' oxygens are surface exposed as in the wild type protein. To clarify this point, we have revised our discussion (page 18).

6. Line 666: If Af1521 alters its favor ADPr acceptor sites (binding sequence) after engineering, from MS data, is there any peptide bound to WT Af1521, but not eAf1521?

Response: As requested by the reviewer we have further investigated our ADP-ribosylome data. The analysis revealed that 5 out of 86 (> 6%) respectively 26 out of 897 (>3%) ADPr peptides were only enriched by WT Af1521 in UT and H₂O₂-treated HeLa cells, respectively. This variation is well within the standard variability accepted between technical replicate MS-experiments, which can be also observed in Suppl. Fig. 2C. Further investigations of the ADP-ribosylated peptides, that are only detected by WT Af1521, revealed comparably low numbers of PSMs, indicating the low abundance of these ADPr sites and no bias in amino acid acceptor sites was observed (S, Y, R were identified with a site localization probability over 95%). We revised the manuscript on page 14 accordingly.

Reviewer #3 (Remarks to the Author):

Nowak and colleagues have successfully engineered a variant of an archaeal ADP-ribose (ADPr) binding protein (Af1521) that displays greater-than 1000-fold binding affinity for ADPr modified protein substrates over the wild-type protein. This construct represents a significant step forward in the development of ADPr-specific enrichment tools and will aid in the characterization of the ADPr-modified proteome, the visualization of ADPr containing proteins via Western blot analysis, and improve the detection of ADPr in vivo using immunofluorescence. Importantly, the development of a tight ADPr binding protein will allow for the identification of novel protein substrates that have thus far evaded detection. This is a very useful and interesting article that will boost various proteomic and biochemical research efforts in the field.

Response: We thank Ian Carter-O'Connell for thoroughly proof-reading our manuscript. We value your suggestions and comments to our manuscript, and we hope we could convincingly answer your questions/comments.

The following are a few minor specific comments regarding the manuscript.

1. In the proteomic analysis, it was noted that variable ADPr acceptor sites were allowed to occur on S, R, K, D, E, and Y. Was there a reason that C was not included in this list? Had the authors based this on previous reports or was it not observed in their preliminary work flow? It may be worth addressing why C was not included.

Response: We agree that investigating all possible amino acid ADP-ribose acceptor site is important, especially for new biological samples (different cell lines, biopsies, different treatments). However, Hendriks et al, 2019 has shown that C, H and T are not very abundant amino acid acceptor sites in untreated and H₂O₂-treated HeLa cells (2). The addition of more amino acids that are likely not present adds computational complexity to the searches that could introduce erroneous results (3). We therefore restricted our initial searches to the acceptor sites S, R, K, D, E, and Y. Nevertheless, as requested by this reviewer we reanalyzed our EThcD data using Mascot including C, H and T and identified only a few peptides that matched the number of peptides potentially modified at D, E or

Y (see Fig. 2D). We thus, did not include these findings in the data of these manuscript. To clarify this aspect, we revised the manuscript accordingly (revised page 13).

2. On line 429, the authors claim that Af1521 "displayed a KD > 1000 times higher than WT Af1521." Should this read lower instead?

Response: Thank you very much for noticing this miswriting. The KD of eAf1521 is 1000 times lower than WT Af1521. We revised the text accordingly (page 13).

3. The analysis of the Y145R mutation is very clear and is supported by the loss of function alteration from Arg to Lys. In this vein, it might be worth demonstrating that the mutation of I141 to either Leu or Val causes the loss of binding affinity for Af1521 and that the proposed phosphate trapping contributes significantly towards the increase in overall affinity.

Response: Due to the basic character of the proposed Leu or Val, we expect that these point mutations would still entrap the ADP-ribose. Therefore, we decided to replace Ile to Gly. Binding assays performed with I144G eAf1521 mutant did not reduced the binding efficiency towards the ADP-ribosylated H2B peptide, indicating that I144 does not contribute to the binding affinity. The new data has been included as new Fig. 1E and the manuscript was revised accordingly (page 14 and 16).

4. In the discussion of the overlap between Af1521 and eAf1521-dependent H2O2-induced proteomes, the authors mention that they observe a 95% overlap between the proteins identified using WT Af1521 and the larger eAf1521 dataset. In the remaining 5% of proteins lost from WT Af1521 to eAf1521, is there any pattern to the lost proteins (i.e. a biological, structural, or sequence-based reason for why eAf1521 didn't capture these proteins)?

Response: Overall, this level of replicate variation is typically observed with all MS-based analyses – 10% variation is well accepted for both technical and biological replicates in MS analyses. 5 out of 86 (> 6%) respectively 26 out of 897 (>3%) ADPr peptides were only enriched by WT Af1521 in UT and H2O2-treated HeLa cells, respectively. Indeed, such variations were observed for the overlap of ADP-ribosylated proteins within the replicates of 5, 10 and 20 mg of starting material (Suppl. Fig. 2G). However, we investigated our ADP-ribosylome data in more detail by comparing the PSM counts of the enriched ADP-ribosylated peptides with the ADP-ribosylated peptides detected in both enrichments. The remaining ADP-ribosylated peptides had lower PSM counts due to their generally lower abundance, but otherwise did not show any other patterns (e.g. amino acid acceptor sites) (revised page 14).

5. The authors claim that the selection strategy that produced eAf1521 did not result in any apparent bias in the selection of ADPr-modified proteins. It might be useful to include a comparison of their current method with previously published proteomic efforts (with antibody, Af1521, or PARG based enrichment) using H2O2-induction to highlight the overlap of the current method with the literature. This would help strengthen their claim regarding the lack of bias.

Response: As suggested by the reviewer, we have performed a comparison with another previously published ADP-ribosylome study using WT Af1521 for enrichment (Hendriks et al. 2019) (new Suppl. Fig. 2F). The ADP-ribosylome data from Hendriks et al. is much deeper due to additional fractionation of the sample into 7 fractions and the protein digestion with Lys-C. Nevertheless, due to the high overlap with other technologies the comparison strengthens our previous claim that we do not observe any bias in H2O2-treated HeLa cells. This aspect was now included and discussed in the revised manuscript (page 14).

6. When the authors attempted their method using decreasing amounts of HeLa lysate, they note that the lowest sample (5 mg) was unable to detect ~20% of the proteins identified at the 20 or 10 mg level. Of the 20% that was lost, was this primarily due to the lower abundance of those proteins in the HeLa cells?

Response: We have performed further analysis comparing the PSM counts of ADP-ribosylated peptides identified in the corresponding 5, 10 and 20 mg lysates enriched by eAf1521. The remaining ADP-ribosylated proteins, which have not been found in the lowest input sample, revealed low numbers in PSM counts due to lower intensities/abundance of the ADP-ribosylated peptides in the sample (page 15).

7. In Fig 4A, the authors present the Western blots comparing WT-Af1521 and eAf1521 based detection. Were these blots taken at the same time with the same exposure? I assumed that they were from the same raw image, but it would be worth explicitly stating this in the figure legend.

Response: Yes, the western blots comparing WT Af1521 and eAf1521 were taken at the same time and with the same exposure. We have clarified this aspect in the revised figure legends (Fig. 4 A and Suppl. Fig. 4 C, page 28, 29 and 30).

Reviewer #4 (Remarks to the Author):

Nowak et al describe the engineering of a new variant of Af1521, a naturally occurring protein that binds ADP-ribosylated proteins, using error-prone PCR, ribosome display, and expression of recombinant protein in E. coli. The affinity of the engineered variant for ADP-ribosylated proteins is 1,000x higher than that of the wild-type Af1521, which makes it a more sensitive detection tool for ADP-ribosylated proteins, e.g., in proteomic applications. The comparison between crystal structures of the wild-type and the variant suggests that the increased affinity is due to a change of orientation of ADP-ribose in the binding site of Af1521.

The strength of this manuscript is in the magnitude of the improvement of the protein being engineered, in the high-resolution crystal structure that allows the interpretation of this improvement at atomic level, and in the demonstration of improved utility of the variant as a detection tool.

The major weaknesses of the manuscript are the gaps in the description of the materials and methods used during ribosome display and screening of the selected variants, the poorly edited text, and an occasional leap to conclusion without supporting evidence.

Response: We thank the reviewer for the critical comments. By revising our manuscript and especially rewrite the materials and methods section, we clarified the raised concerns (see revised page 5, 6 and 12).

Also, I wish they had used site-directed mutagenesis of the engineered variant to determine which of the 9 mutations in the variant are essential for the improved binding phenotype, and which are incidental.

Response: We have mutagenized the two point mutations of the eAf1521 in the ADPr binding region to phenocopy the ADP-ribose binding region of WT Af1521 and vice versa and repeated the binding assays with the ADP-ribosylated H2B peptide. These experiments revealed that the amino acid substitutions K35E and Y145R are sufficient to convert WT Af1521 to a mutant with comparably increased affinity towards ADPr. Reverting the two mutants in eAf1521 also reduced the affinity to ADPr in eAf1521, strongly suggesting that only the two mutated residues are responsible for the increase affinity to ADPr. We have included these additional experiments (new Fig. 1E) and revised the text accordingly (page 14 and 18).

Specific comments follow:

- Line 62: “Several different proteins able to bind poly-ADPr (e.g. PAR binding motif (PBM)), two consecutive ADPr units (e.g. PAR-binding zinc finger (PBZ)) or iso-ADPr (e.g. WWE domain) have been described (i.e. readers).”Missing reference. Also, it’s not clear what “readers”refers to.

Response: As requested by this reviewer, we have included a reference and explained the term readers (revised on page 4).

- Line 77: “our understanding of the cellular processes regulated by ADP-ribosylation had remained very modest for a long time due to the limited tools available (e.g. antibodies).”What traditional tools were “limited,”and in what way were they “limited”? Antibodies have done very well in other applications. How have they failed in this field?

Response: ADP-ribosylation was traditionally studied and identified in vitro by the incorporation of radioactive NAD⁺ or NAD⁺-analogs. For a long time, only antibodies against poly-ADP-ribosylation (PAR) were available, limiting the analysis by immunoblotting or immunofluorescence for PARylated protein. Only recently the ADP-ribose binding domains, such as the macro domain Af1521, fused to Fc-fragment or an antibody that can detect mono-ADP-ribosylation have become available (revised on page 3).

- Line 110: “The randomization occurs either at a low rate by the intrinsic error rate of the polymerase used but can be enhanced using error-prone PCR amplification methods (51), DNA shuffling (52), or both.” There are additional

and very common sequence-diversification methods, most of which take advantage of in vitro oligonucleotide synthesis, using mixtures of nucleotides or of codons. Reference (52) is 25 years old.

Response: We are aware that different methods exist and that they do not come without any bias. However, the strategy of using enhanced error-prone PCR using nucleotide analogs have been validated in our lab and also have led to the identification of mutations that increase affinity in the study presented here. There are numerous publications of recent years where this strategy has successfully been applied. We prefer for the readers not to neglect the original publication, but we could also include some more recent publications (e.g. Dreier et al. 2011, J. Mol. Biol. 405, 410-426; Luginbühl et al. 2006, J Mol Biol 363, 75-97; Ahmad et al. 2016, Sci. Rep. 6, 28922; Stefan et al., 2011, J. Mol. Biol. 413, 826-843). We have clarified this aspect in the revised manuscript (revised on page 4).

•Line 139: The lack of detail about how the ribosome-display selection was performed is puzzling. For example: “After error-prone PCR, the mutagenized Af1521 constructs were transcribed and translated in vitro as described earlier (53). For the first selection round, 200 nM of biotinylated ADP-ribosylated H2B peptide (Fig. S1) was immobilized on a precoated streptavidin plate. The ribosome complexes displaying in vitro translated mutagenized Af1521 domains were added to the wells and incubated for 1 hr at 4 °C, and subsequently each well was immediately washed twice and additionally 4 times for 2 min. Afterwards a reverse transcription was performed using the recovered mRNA and subsequently amplified by PCR using the scaffold-specific primers Af1521_pRDV_fwd_2 5’-gacaaaggatccATGGAACGGCGTAC-3’ and Af1521_pRDV_Eco_rev 5’-CTTTGAGAGGAGTCTTgaattcgga-3’.” The sequences of the PCR primers are given, but very little is revealed about the selection protocol. What was the volume at each round? What was the source of the streptavidin plates? The amount of RNA/DNA? The buffer? The authors state that transcription and translation followed the protocol in (53), but don’t tell us anything about the selection protocol. If they did follow the selection protocol in reference (53) exactly, they need to state that. If they followed it mostly but made some changes, the changes need to be specified.

Response: We apologize for the fact that the selection process was described rudimentary. Since it is a very well-established process in our laboratory and was published before, we did not give this technology sufficient emphasis which we would like to correct now: For the selection of Af1521 mutants with improved affinity, the Af1521 cDNA was cloned into the pRDV plasmid backbone using the restriction enzymes BamHI and EcoRI. In order to introduce random mutations error prone PCR was applied using the pRDV-specific primers T7B and tolAk using either 0, 3, 6 and 10 µM of the nucleotide analogs dPTP and 8-oxo-dGTP each and Platinum Taq Polymerase (Invitrogen) as previously described (4). For the PCR using the outer primers the following conditions applied: initial denaturation 3 min at 95°C, for amplification 40 cycles with 30 sec at 95°C, 30 sec at 50°C and 1 min at 72°C followed by a final extension at 72°C for 5 min. The four PCR reactions resulting in different mutational loads were pooled and used as template for the in vitro transcription reaction using T7 RNA polymerase (Fermentas) and a home-made transcription buffer exactly as described (4). Resulting RNA was purified using Illustra Microspin G-50 columns (GE). Ten µg of purified RNA were used for the in vitro translation reaction in a volume of 110 µl containing home-made S30 extract and premixZ/methionine. In vitro translation was performed at 37°C for 15 min before the reaction was stopped as described (4). From the stopped in vitro translation reaction 4 times 100 µl were used for the first selection round, only 2 times 100 µl for successive rounds which were pooled after the elution step. To remove unspecific binders a pre-selection step was included in round 2 and 3, but omitted from the initial selection round using a BSA-blocked streptavidin-coated (Immunopure, Pierce) well of a 96 well Maxisorp plate (Nunc) for 30 min at 4°C. The reaction was then directly transferred to a fresh well additionally containing the biotinylated, streptavidin-immobilized ADP-ribosylated H2B peptide. The binding reaction was performed at 4°C for 1 hr. In the initial round 200 nM of peptide was used, while in rounds 2 the target concentration was reduced to 100 nM and in round 3 and 4 to 20 nM, respectively. The selection wells were washed 6 times (2 fast washes followed by 4 washing steps with a 2 min incubation) with ice-cold 300 µl WTB buffer (50 mM Tris-acetate, 150 mM NaCl, 50 mM Magnesium acetate, 0.05 % Tween-20, pH 7.5), For all other rounds the washing was prolonged to two fast washes and four washing steps of 10 min incubation each. Elution was performed by addition of twice 100 µl EB buffer (50 mM Tris-acetate, 150 mM NaCl, 25 mM EDTA) followed by RNA purification and DNaseI treatment using the HighPure RNA isolation kit (Roche) as described previously (4). Reverse transcription was performed essentially as described (4) with the exception that the scaffold-specific primer Af1521_pRDV_fwd_2 5’-gacaaaggatccATGGAACGGCGTAC-3’ was used. PCR was performed using the scaffold-specific primers Af1521_pRDV_fwd_2 5’-gacaaaggatccATGGAACGGCGTAC-3’ and

Af1521_pRDV_Eco_rev 5'-CTTTGAGAGGAGTCTTgaattcgga-3' with an annealing temperature of 50°C and 35 cycles using Vent polymerase (NEB). Respective bands were gelpurified using the Gelpurification Kit (QIAGEN) and cDNA fragments were digested with BamHI and EcoRI followed by a clean-up using the PCR purification kit (QIAGEN) and ligated into pRDV. The resulting enriched pool served as template for the next round of selection using error-prone PCR and the primers T7B and TolAk. In total four rounds of selection were performed with increasing stringency lowering target protein concentration and increasing washing time. In round three and four an off-rate selection was implemented as described previously (4). In addition, in round 4 the selection was performed in solution using MyOne T1 streptavidin coated magnetic beads (Thermo Scientific) as previously described (4). While for round 3 error-prone PCR was applied this step was omitted prior to round 4 which served solely to enrich for highly specific H2B binders. Briefly, for round three a competition with 1000-fold excess of competitor was performed. For round 4 20 nM of biotinylated ADP-ribosylated H2B peptide in the presence of 20 μM of ADP-ribosylated ARTD10cat (GST-fusion protein expressing the catalytic domain of ARTD10/PARP10 that is able to catalyze auto-mono-ADP-ribosylation) was used and incubated for 1 hr to increase the specificity towards the ADPr. A 30 min capture of Af1521 mutants that still bound to the biotinylated peptide was performed using streptavidin-coated magnetic beads as previously described (4). Af1521 mutants still binding to the biotinylated H2B peptide were eluted with EB buffer followed by RNA purification and RT-PCR. We revised the text accordingly (page 5, 6 and 12).

•Line 157: “As competitor 20 nM of automodified ARTD10cat was used to increase the specificity towards the ADP-ribosylated H2B peptide.” What is ARTD10cat? What was the competition protocol? How did that increase the specificity?

Response: We have answered the reasoning and technical details above and hope that this is sufficient (see response to point above). Only the last pool of the selection (round 4) was analyzed by ELISA. It is therefore not possible to make a statement of how the specificity improved by the off-rate selection using ADP-ribosylated ARTD10.

•Line 161: “After 96 well expression of the mutagenized Af1521 domains and an ELISA was performed against the ADP-ribosylated H2B peptide (biotinylated and immobilized on streptavidin plates) using 10 μL of bacterial lysate. The binding was detected using the MRGS-H6-tag (45).” How did the variants enriched during the selection make it from line 160 to line 161? How many wells or 96-well plates of variants were expressed? What reagents and conditions were used in the ELISA? At what stage were the clones sequenced? How many different variants of Af1521 were purified and characterized? How did the final lead compare to the other variants considered?

Response: The enriched pool of Af1521 mutants after the fourth round of ribosome display selection was subcloned via BamHI and PstI into pDST67 (5) a pQE30 (QIAGEN) derivative as fusions with a N-terminal MRGS(H)₆ tag. After transformation of E. coli XL-1 Blue 92 single colonies were picked from selection plates and expressed in a 96 well format in 2xYT media containing 100 μg/ml ampicillin and 1% glucose for 4 hrs at 37°C after induction using 0.5 mM IPTG. Cells were harvested by centrifugation for 10 min at 4000 rpm and cells lysed by addition of 50 μl B-PERII cell lysis buffer (Pierce) as previously described (4). The lysate was diluted with 950 μl PBS-TB (PBS: 137 mM NaCl, 30 mM KCl, 80 mM Na₂HPO₄, 15 mM KH₂PO₄, pH 7.4 containing 0.1 (v/v) Tween20 and 0.2% (w/v) BSA) and cleared by centrifugation for 10 min at 4000 rpm. In order to identify initial hits an ELISA was performed against the biotinylated and ADP-ribosylated H2B peptide and unmodified peptide as control. In brief, 96 well MaxiSorp plates (Nunc) were coated with 100 μl 66 nM Immunopure streptavidin (Pierce) in PBS overnight at 4°C. After washing twice with PBS, the wells were blocked with PBS/0.5% BSA for 1 hr at room temperature (RT). The biotinylated peptides were immobilized at a concentration of 100 nM in PBS-TB at RT for 1 hr. After three washes with PBS-T 100 μl of diluted lysate (90 μl PBS-TB plus 10 μl lysate) was added to the target-coated wells and incubated at RT for 1 hr. Following three washes with PBS-T peptide-bound Af1521 mutants were detected using a mouse- anti-RGS(H)₄ primary antibody (QIAGEN) followed by a goat-anti-mouse secondary antibody coupled to alkaline phosphatase (Sigma). After addition of the substrate (pNPP) absorbance at 405 nm was determined as previously described (4). 10 clones were randomly selected for sequencing. We have included this information in the revised manuscript (page 6 and 12). The two last questions of the reviewer’s comment are answered in the next response.

•Line 411: “Many clones showed a significant binding signal to the ADP-ribosylated H2B peptides over the signal of the unmodified peptide (data not shown).” This data needs to be shown, at least in supplementary materials. How many clones were tested? How many bound? How is “significant” binding defined?

Response: After the 4th ribosome display selection iteration, the pools were subcloned into an E. coli expression vector for subsequent analysis of the mutated Af1521 domains by ELISA, using the modified H2B peptides and its unmodified counterpart (6). 88 out of 92 clones (> 95%) displayed specific and significant ADP-ribosylated peptide binding properties (binding signals > 0.5 absorbance units (AU)) compared to the unmodified peptide (new Suppl. Fig. 1C). Due to the large number of positive binders, we randomly chose 10 clones and subcloned them as GST-fusion proteins. After expression and purification, increasingly stringent pull-down assays using unmodified and ADP-ribosylated H2B peptides were performed to further characterize the candidate binders (Fig. 1A, new Suppl. Fig. 1D). While binding of WT Af1521 to the ADP-ribosylated peptide was compromised at salt concentrations of 200 mM and lost at 400 mM, we observed that only one of the tested macro domain candidates was still able to bind the modified peptide very well in the presence of 400 mM NaCl, which we termed eAf1521. We revised the text accordingly (page 12).

•Lines 464-472 describe a limited attempt to explore the contribution of a salt bridge between two mutated residues to the increase in affinity for ADPr. Only a single mutation from R to K was made, and the resulting reduction in affinity led the authors to attribute the high affinity of the engineered variant to the removed guanidine group. In the absence of the crystal structure of this RK mutant, we don't know what effects on the structure and function this mutation had. A more extensive mutagenesis experiment may be able to pinpoint the role of this salt bridge, and the roles of the other 7 mutations, in improving the affinity for ADP-ribosylated proteins.

Response: To investigate the contribution of the salt bridge generated by the two amino acid substitutions K35E and Y145R in eAf1521, we performed site-directed mutagenesis to phenocopy the two residues in WT Af1521 and vice versa for eAf1521. Repeating the binding assays with the ADP-ribosylated H2B peptide revealed that these point mutations are mainly responsible for the observed increased affinity towards ADPr of eAf1521. We have included these additional experiments (new Fig. 1E) and revised the text accordingly (page 14 and 18).

•Line 672: The authors argue that the engineered detection reagent is unlikely to show a bias for the peptide that it was selected against given that it has pulled down unrelated proteins also. Then they follow with, “Nevertheless, to overcome any potential bias that eAf1521 may exhibit towards certain ADPr acceptor sites, we would recommend performing the enrichment of ADP-ribosylated peptides with both WT Af1521 and eAf1521 in parallel or in tandem.” This recommendation is surprising given that the work described was directed towards identifying a more sensitive and thus more useful detection reagent, and given that this goal appears to have been achieved. Instead of continuing to rely on the older and less sensitive reagent, let's test both reagents against a range of different ADP-ribosylated proteins and compare their affinities, in vitro, to determine whether there is such a bias

Response: We agree with the review, that this is a very interesting aspect to study. However, we do currently not have tools available to properly investigate a potential bias, since it is not yet possible to generate peptides carrying ADP-ribose at different amino acid acceptor sites. In the past, we tried to modify target proteins on specific amino acid acceptor sites by in vitro ADP-ribosylation assays, but subsequent MS-analysis revealed that these assays are very promiscuous. Several amino acid acceptor sites were modified without any particular pattern (e.g. motif). Moreover, using cell culture systems to address the bias is not possible due to high complexity of different amino acid acceptor sites and targets. In the result part of our manuscript we mentioned that eAf1521 enriches four selected S-ADPr peptides (e.g. HNRNPA1, SSGPYGGGGQYFAKPR,) to a higher extent while the binding efficiency towards two selected R-ADPr peptides (e.g. PDIA3, RLPEYEAAATR) compared to WT Af1521 remained unchanged (page 16, Fig. 3 B and C). The observed enrichment might also originate from the nature of the glycosidic linkage to the ADP-ribose (O- versus N-glycosidic bond). We have revised the manuscript accordingly (revised on page 19).

We are convinced that researchers investigating the role of ARTD1 or other ARTs, mediating S-ADPr can use eAf1521 as a great tool for their research. However, we aimed to raise awareness that this tool might have limitations for different biological research questions e.g. investigating ARTs that catalyze ADP-ribosylation on other amino acid acceptor sites. But this holds not only true for eAf1521 but also for all other detection and enrichment tools in our field including WT Af1521 and potential mono-ADP-ribosylation antibodies.

•The manuscript is difficult to read and would benefit from careful editing for grammar and style.

Response: We hope by addressing your comments we could improve the legibility of our manuscript. We hope we could raise your interest in our research field by answering your questions and concerns.

Reference mentioned above:

1. Andersson, A., Bluwstein, A., Kumar, N., Teloni, F., Traenkle, J., Baudis, M., Altmeyer, M. and Hottiger, M.O. (2016) PKC α and HMGB1 antagonistically control hydrogen peroxide-induced poly-ADP-ribose formation. *Nucleic Acids Res*, **44**, 7630-7645.
2. Hendriks, I.A., Larsen, S.C. and Nielsen, M.L. (2019) An Advanced Strategy for Comprehensive Profiling of ADP-ribosylation Sites Using Mass Spectrometry-based Proteomics. *Mol Cell Proteomics*, **18**, 1010-1026.
3. Leslie Pedrioli, D.M., Leutert, M., Bilan, V., Nowak, K., Gunasekera, K., Ferrari, E., Imhof, R., Malmstrom, L. and Hottiger, M.O. (2018) Comprehensive ADP-ribosylome analysis identifies tyrosine as an ADP-ribose acceptor site. *EMBO Rep*, **19**.
4. Dreier, B. and Plückthun, A. (2012) Rapid selection of high-affinity binders using ribosome display. *Methods Mol Biol*, **805**, 261-286.
5. Huber, T., Steiner, D., Rothlisberger, D. and Plückthun, A. (2007) In vitro selection and characterization of DARPins and Fab fragments for the co-crystallization of membrane proteins: The Na⁽⁺⁾-citrate symporter CitS as an example. *J Struct Biol*, **159**, 206-221.
6. Binz, H.K., Amstutz, P., Kohl, A., Stumpp, M.T., Briand, C., Forrer, P., Grutter, M.G. and Plückthun, A. (2004) High-affinity binders selected from designed ankyrin repeat protein libraries. *Nat Biotechnol*, **22**, 575-582.

REVIEWERS' COMMENTS:

Reviewer #1 (Remarks to the Author):

The authors have now provided additional data and discussions addressing my concerns. The manuscript is ready to be published.

Reviewer #3 (Remarks to the Author):

Nowak and colleagues have adequately addressed all of the points raised in the prior review of their manuscript entitled: "Engineering Af1521 improves ADP-ribose binding and identification of ADP-ribosylated proteins." Their manuscript has been revised accordingly and the novel eAf1521 construct they detail will be of significant aid in the field.

Reviewer #4 (Remarks to the Author):

I'd like to thank the authors for addressing my comments and requests for additional information thoroughly.